# Few Contrastive Attention Heads Enable Visual Grounding in Large Vision-Language Models

## Abstract

Visual grounding aims to localize image regions corresponding to natural language expressions. While recent Large Vision-Language Models (LVLMs) have shown impressive multimodal understanding capabilities, their application to visual grounding typically requires fine-tuning and architectural modifications. This requirement, however, can be ignored, considering that text and images tend to have similar feature representations that appear to be approximately linearly disentangled, enabling cleaner extraction of spatial information from LVLMs without any task-specific training. From this viewpoint, we propose an attention-head discovery framework that requires zero labeled grounding samples and no architectural modifications, and identifies discriminative localization heads without manual inspection. Through dual prompting with target and contrastive descriptions, we compute differential residual representations and project them through attention head output matrices to measure per-head spatial contributions via four complementary scores. By aggregating signals using importance-weighted query difference scores from only the top-10 attention heads, we outperform training-free non-LVLM baseline by up to 27.95% on RefCOCO, 21.93% on RefCOCO+, and 8.40% on RefCOCOg. Our method outperforms LVLM baseline by up to 8.04% on RefCOCO without requiring ground-truth category labels.

## 1 Introduction

Visual grounding is a multimodal reasoning task that aims to associate natural language descriptions with their corresponding visual entities in an image. Given an image and a textual query, the objective is to comprehend the semantic content of the language and accurately identify and localize the visual region that matches the description Mao et al. (2016); Yu et al. (2016); Sadhu et al. (2019).

Recently, this vision–language task, which inherently requires a deep understanding of the relationships between visual content and natural language, has witnessed substantial progress driven by the emergence of powerful Large Vision–Language Models (LVLMs) Li et al. (2023); Liu et al. (2024a; 2023); Tong et al. (2024). However, since LVLMs are primarily designed for text generation, directly applying them as vision–language models for identifying and localizing objects within images—that is, for visual grounding—poses significant technical challenges. Consequently, existing LVLM-based visual grounding approaches typically rely on explicit fine-tuning of LVLMs using additional visual grounding datasets, along with modifications to model components to support the generation of spatial outputs such as bounding boxes Wang et al. (2024b); You et al. (2023) or segmentation masks Rasheed et al. (2024); Lai et al. (2024). This requirement, however, can be relaxed, considering that text and image features tend to share similar representations that are approximately linearly disentangled—enabling cleaner extraction of spatial information from LVLMs without any task-specific training Merullo et al. (2022).

Despite the promising integration of LVLMs in prior visual grounding studies, this raises a fundamental question:

> *Can we exploit the linear disentanglement of text-image features to extract cleaner spatial information from LVLMs without any labeled grounding data, relying solely on the model's own spatial representations at inference time?*

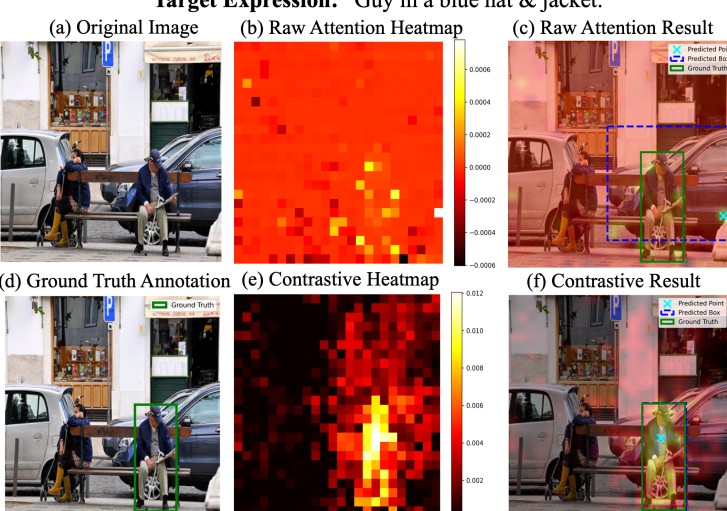

Figure 1: Visualization of text-to-image attention maps from LLaVA-1.5-7B Liu et al. (2023). While the raw attention heatmap (b) initially appears unfocused and spreads attention across the entire image, the contrastive head selection method (e) discovers that the information of the objects is encoded in specific attention heads, but is quite noisy, which we can remove through our method to precisely localize objects when properly identified and aggregated.

More specifically, we investigate whether LVLMs encode spatial grounding by attending to image regions corresponding to textual descriptions. Text-to-image attention maps are commonly analyzed in vision transformers and multimodal models Gao et al. (2021); Cao et al. (2023); Tang et al. (2023), however, naive averaging across all attention heads conflates diverse head behaviors, yielding diffuse heat maps that fail to capture meaningful spatial structure, as demonstrated in Fig. 1.

This observation raises a critical question: if LVLMs genuinely encode spatial understanding in their internal representations, why do their attention maps appear so unfocused? Our key insight is that localization signals reside in a small subset of specialized attention heads, which can be discovered through contrastive reasoning, and remain obscured in averaged attention maps.

We present an efficient and training-free framework for visual localization where attention differentials computed between target and contrastive forward passes are combined to generate spatial probability distributions for referred objects. Remarkably, selecting merely ten attention heads proves sufficient for accurate object localization, indicating their inherent specialization toward discriminative visual features. Unlike conventional fine-tuning methods, our approach operates without additional training, removing the need for task-specific model adaptation in referring expression scenarios.

We show that spatial grounding signals are linearly decodable in the residual stream, enabling automatic discovery of specialized localization heads via four complementary scoring functions — without any labeled data or model modification. This reveals that models inherently encode directional understanding extractable directly from token space: when localizing objects in specific positions, corresponding directional tokens (e.g., "left", "right") naturally emerge with higher activation values between them. To our knowledge, no existing training-free method addresses attention-based score setting for visual localization in large-scale vision-language architectures.

Our contributions can be summarized as follows:

- We discover that frozen LVLMs encode intrinsic spatial grounding signals in a small subset of specialized attention heads. We show that these signals are linearly separable in the residual stream, consistent with the linear representation hypothesis, and can be extracted without any fine-tuning or labeled grounding data.

- We propose an automatic contrastive descriptor generation mechanism that derives spatially-opposing region descriptors directly from the model's output logits over a predefined spatial vocabulary, eliminating the need for manual contrastive prompt design.

- We introduce four complementary scoring functions — attribution via output projection, image patch attribution, attention difference, and query difference scores — that jointly identify the attention heads most critical for visual grounding, enabling importance-weighted heatmap aggregation without any supervision.

- We evaluate our method across multiple LVLMs of varying scales on standard referring expression benchmarks, demonstrating competitive performance against existing training-free approaches.

## 2 Related Work

**Visual Grounding.** Visual grounding aims to localize image regions referred to by natural language expressions Raghavi Chandu et al. (2021), with Referring Expression Comprehension (REC) Mao et al. (2016); Yu et al. (2016) focusing on bounding-box localization. Early transformer-based approaches such as MDETR Kamath et al. (2021) and SeqTR Zhu et al. (2022) established cross-modal alignment as a core design principle, later scaled by GLIP-L Li et al. (2022), G-DINO Liu et al. (2024b), ONE-PEACE Wang et al. (2023), and UNINEXT Lin et al. (2023). These methods achieve strong grounding accuracy but require substantial task-specific training on grounding datasets.

**LVLMs for Visual Grounding.** Recent work extends large vision-language models for grounding by incorporating region-level localization components, typically requiring fine-tuning on grounding-specific data Wu et al. (2025). Representative approaches include Shikra Chen et al. (2023), Ferret You et al. (2023), and CogVLM Wang et al. (2024b), which augment general-purpose VLMs with spatial output heads for bounding box or mask prediction. SpatialVLM Chen et al. (2024) further models explicit spatial relations, while GLaMM Rasheed et al. (2024) enables multi-granular scene-to-region understanding. Although effective, all these methods couple grounding capability to task-specific supervision Voita et al. (2019); Clark et al. (2019), motivating the exploration of training-free alternatives.

**Training-Free Visual Grounding.** Training-free approaches avoid task-specific supervision by repurposing foundation models for grounding. CLIP-based methods Radford et al. (2021); Kim et al. (2025) score region proposals by text-image similarity, while diffusion-based approaches Rombach et al. (2022) exploit cross-attention maps as spatial localization signals. Representative methods include ReCLIP Subramanian et al. (2022), GroundVLP Shen et al. (2024), and CPT-Blk Yao et al. (2024). However, these approaches rely on contrastive vision encoders or generative models rather than the internal representations of autoregressive LVLMs. Training-free grounding that exploits the attention structure of frozen LVLMs remains largely unexplored.

**Attention Analysis in Transformers.** A growing body of work studies how attention heads in transformers encode semantic and spatial functions. The linear representation hypothesis posits that semantic concepts are encoded as linearly disentangled directions in the residual stream Elhage et al. (2022); Park et al. (2023); Saurez et al. (2026), a principle supported empirically by linear probes Alain & Bengio (2016) and sparse autoencoders Ng (2011). In vision-language models, attention maps have been analyzed for text-to-image correspondence Gao et al. (2021); Tang et al. (2023), but naive aggregation across all heads produces diffuse heatmaps that obscure spatial structure. Our work builds on these insights to identify specialized localization heads through contrastive residual analysis, without requiring any labeled data. Kang et al. (2025) demonstrated that a small subset of attention heads in LVLMs is sufficient for visual grounding, identifying these heads by computing spatial-entropy statistics over 1,000 labeled RefCOCO samples.

While effective, this approach couples head selection to a specific dataset distribution, requiring re-annotation for new domains. Our work addresses the same question — *which heads matter for grounding?* — but answers it without any labeled data, discovering localization heads automatically from the model's own

spatial representations at inference time via contrastive prompting. The comparison with the other methods is presented in Table 1.

Table 1: Positioning against head-selection approaches. "Selection signal" = what identifies the heads; "Granularity" = whether the head set adapts to each input; "Causal validation" = whether the selected heads are verified via intervention (ablation / patching) rather than correlation alone.

| Method | Selection signal | Labels / training | Granularity | New domain needs | Causal validation |
|---|---|---|---|---|---|
| Voita et al. (2019); Clark et al. (2019) | task-supervised pruning / probing | task labels | fixed | re-analysis | partial (pruning) |
| F-LMM (Wu et al., 2025) | frozen attention + trained refiner | grounding data (training) | fixed (learned) | retraining refiner | — |
| Kang et al. (2025) | attention sum + spatial entropy stats. | 1,000 samples + $\tau$ | fixed per model | re-calibration (1,000 samples) | — |
| **Ours** | **contrastive residual / attention / query analysis** | **none** | **per-sample** | **none** | **ablation + patching (App. A.1)** |

## 3 Preliminaries

### 3.1 Vision-Language Model Architecture

LVLMs consist of three primary components: a vision encoder, a projection module, and a large language model. Given an input image $\mathbf{X}_v$, the vision encoder processes the image and the projector maps visual features into the language model embedding space, producing visual token embeddings $\mathbf{Z}_v \in \mathbb{R}^{P^2 \times d}$, where $P^2$ represents the number of flattened image patch tokens and $d$ denotes the hidden dimension. Similarly, input text $\mathbf{X}_t$ is converted into token embeddings $\mathbf{Z}_t \in \mathbb{R}^{L \times d}$, where $L$ is the number of text tokens. These embeddings are concatenated as $\mathbf{Z}_0 = [\mathbf{Z}_v; \mathbf{Z}_t] \in \mathbb{R}^{(P^2+L) \times d}$ and fed into the language model.

### 3.2 Multi-Head Self-Attention

The input embeddings $\mathbf{Z}_0$ pass through multiple decoder layers, each comprising multi-head self-attention and feed-forward modules. In layer $\ell$ and attention head $h$, the hidden state from layer $\mathbf{Z}_{\ell-1}$ is projected into query $\mathbf{Q}_{\ell,h}$, key $\mathbf{K}_{\ell,h}$, and value representations $\mathbf{V}_{\ell,h} \in \mathbb{R}^{(P^2+L) \times d_h}$ with the head dimension $d_h$. The attention mechanism computes:

$$\mathbf{A}_{\ell,h} = \mathrm{softmax}\left(\frac{\mathbf{Q}_{\ell,h}\mathbf{K}_{\ell,h}^{\top}}{\sqrt{d_h}}\right) \tag{1}$$

The output of the attention head is given by:

$$\mathbf{O}_{\ell,h} = \mathbf{A}_{\ell,h}\mathbf{V}_{\ell,h}. \tag{2}$$

The attention weights reflect the similarity between query and key vectors, capturing the interaction between different tokens in the sequence.

### 3.3 Image-Text Interaction Analysis

Given the auto-regressive nature of language model decoding, information flows from preceding tokens to subsequent ones, making the final token's representation encompass the entire sentence context. We denote the query vector of the last input text token at layer $\ell$, head $h$ as $\mathbf{q}_{\mathrm{txt}} \in \mathbb{R}^{d_h}$, which serves as a representative

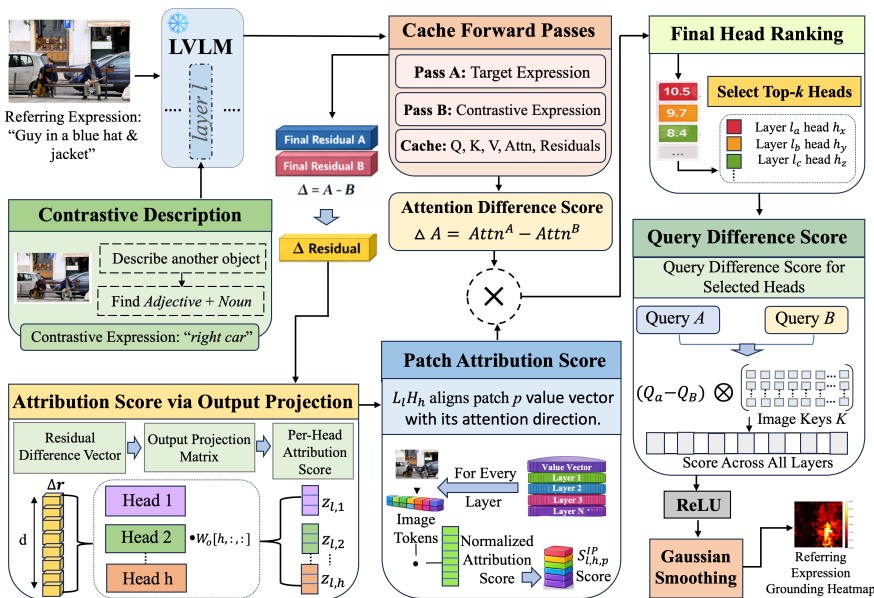

Figure 2: Four complementary scores for identifying attention heads that capture spatial relationships in referring expressions. By integrating these scores into a head ranking mechanism, we systematically identify heads most aligned with spatial grounding, which are used to generate accurate localization heatmaps.

query for the complete text expression. To analyze image-text interactions, we examine attention weights where the query is $\mathbf{q}_{\text{txt}}$ and keys correspond to all tokens:

$$\mathbf{a}_{\ell,h} = \text{softmax}\left(\frac{\mathbf{q}_{\text{txt}}\mathbf{K}_{\ell,h}^{\top}}{\sqrt{d_h}}\right) \in \mathbb{R}^{P^2+L} \tag{3}$$

We focus on the components corresponding to image patches, denoted as $\mathbf{a}_{\ell,h}^{\text{img}} = \mathbf{a}_{\ell,h}[1:P^2] \in \mathbb{R}^{P^2}$, representing attention weights over the visual tokens. These attention weights indicate which image regions the model focuses on when processing the text expression. For notational brevity, we denote layer $\ell$ head $h$ as $\mathcal{L}_\ell\mathcal{H}_h$ throughout this paper (e.g., $\mathcal{L}_5\mathcal{H}_3$ refers to the third head in the fifth layer).

## 4 Proposed Framework

Our approach generates spatial attention heatmaps through contrastive analysis of vision-language model representations. Our training-free framework automatically discovers discriminative localization heads without requiring manual inspection. This relies on the assumption that transformers encode semantic functions as linearly disentangled vectors — the same principle underlying linear probes Alain & Bengio (2016) and sparse autoencoders Ng (2011), and consistent with the linear representation hypothesis Saurez et al. (2026); Park et al. (2023); Elhage et al. (2022). Given this linearity, we approximate spatial importance by measuring the linear contribution each attention head makes to the final output when processing complementary spatial descriptions, as shown in Fig. 2.

### 4.1 Contrastive Region Selection via Dual Prompting

We perform a forward pass with prompt template $p =$ "USER: ⟨image⟩ [query] ASSISTANT:" where the query is "Describe the $e$", where $e$ denotes the referring expression. From the next-token logits at the final prompt position, we rank spatial adjectives $\mathcal{A} = \{\text{left, right, top, bottom}\}$ in descending order and nouns $\mathcal{N}$ in ascending order, letting $a_1, a_2$ be the top-two adjectives and $s_1, s_2$ their scores. When the margin

between the top two adjective scores is less than 1 logit unit (i.e., $s_1 - s_2 < 1$), we combine them into a compound descriptor $o^* = a_1\text{-}a_2$; otherwise $o^* = a_1$. If no adjective scores are available, $o^*$ falls back to the lowest-scoring noun in $\mathcal{N}$. ==Notably, $a_1, a_2$ are extracted from the model's logits over $\mathcal{A}$, not from the surface form of $e$; the method thus imposes no positional requirement on $e$, as even appearance-only queries (e.g., "red car") yield a dominant spatial adjective from the model's implicit localization, which $\phi$ then inverts.== Features extracted from the image implicitly encode spatial information, as different attention heads tend to focus on different regions of the image. This spatial sensitivity allows us to separate features corresponding to spatial locations, which can then be used to construct more informative contrastive prompts. No dataset is used to obtain the different attention heads; instead, they are obtained individually for each datapoint to give more weight to the training-free aspect.

Using the established prompt template, we generate two inputs with different queries as passes: (A) **Target**:"Describe with detail the $e$", focusing on the target region. (B) **Contrastive**: "Describe the scene, specially around the $\phi(o^*)$ area; and, excluding the $e$", where $\phi$ applies spatial flip mapping left $\leftrightarrow$ right, top $\leftrightarrow$ bottom. When both directions appear simultaneously in $\phi(o^*)$ after flipping (e.g., "right-left"), the descriptor is replaced with "center", as shown in Fig. 3.

Each sequence contains $\mathbf{x} = [\mathbf{x}_{\text{prompt}}, \mathbf{x}_{\text{image}}, \mathbf{x}_{\text{query}}]$ where $\mathbf{x}_{\text{image}} \in \mathbb{R}^{576}$ represents $24 \times 24$ image patches from the $336 \times 336$ resized input.

### 4.2 Attribution Score via Output Projection

We first compute the difference in final output residuals between the two passes:

$$\Delta \mathbf{r} = \mathbf{r}_{l^*}^A - \mathbf{r}_{l^*}^B \in \mathbb{R}^d \tag{4}$$

where $l^*$ is the last layer and $\mathbf{r}_{l^*}^{(A/B)}$ is the residual stream vector from that layer. Since communicable semantic features are encoded as consistent directions in residual space, and a single such direction spans the entire feature subspace, the contrastive pass $B$ cancels features shared between both descriptions. Consequently, $\Delta \mathbf{r}$ lies predominantly in the subspace of the target feature, isolating its direction. To measure each head's causal contribution, we project $\Delta \mathbf{r}$ as:

$$\mathbf{z}_{\ell,h} = \mathbf{W}_O^{(\ell)}[h, :, :]^\top \Delta \mathbf{r} \in \mathbb{R}^{d_h} \tag{5}$$

Geometrically, $\Delta \mathbf{r}$ isolates the differential features between these two passes, highlighting the feature representations from the target pass. This projection allows to measure better the importance of the components given that we are removing all the noise from the residual stream. The contrastive prompting isolates spatial features by canceling shared non-spatial semantics.

### 4.3 Image Patch Attribution Score

To measure each patch's potential contribution independent of attention weights, we compute:

$$s_{\ell,h,p}^{\text{IP}} = \hat{\mathbf{z}}_{\ell,h} \cdot \mathbf{v}_p^{(\ell,h)}, \quad \text{where} \quad \hat{\mathbf{z}}_{\ell,h} = \frac{\mathbf{z}_{\ell,h}}{\|\mathbf{z}_{\ell,h}\|} \tag{6}$$

This measures the magnitude of each patch's contribution when projected through $\mathbf{W}_O$: ==patches whose value vectors $v_p^{(\ell,h)} \in \mathbb{R}^{d_h}$ for image patch $p$ are aligned to $\hat{\mathbf{z}}_{\ell,h}$ would maximally increase the differential residual toward the target representation if attended to.== The score thus reveals which patches contain the spatial features that the head's circuit uses to distinguish target from contrastive descriptions.

### 4.4 Attention Difference Score

We further compute the difference in attention weights between the two passes:

$$\Delta \mathbf{a}_{\ell,h} = \mathbf{a}_{\ell,h}^{\text{img},(A)} - \mathbf{a}_{\ell,h}^{\text{img},(B)} \in \mathbb{R}^{P^2} \tag{7}$$

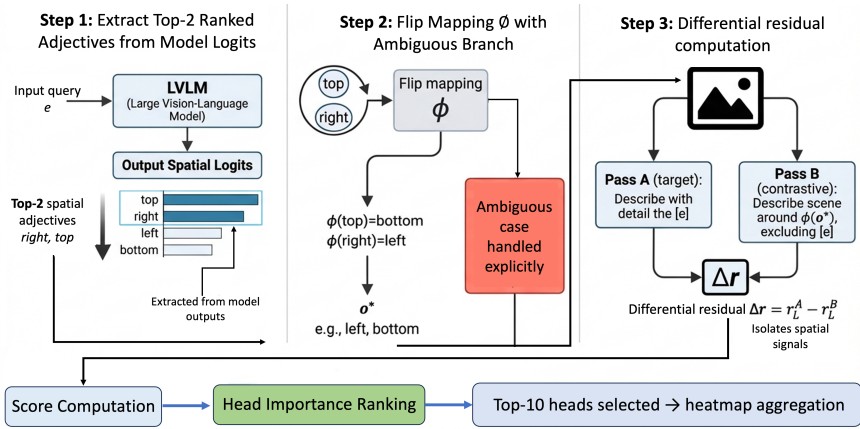

Figure 3: Contrastive prompt generation pipeline. **Step 1:** Logit scores over spatial adjectives $\mathcal{A} = \{left, right, top, bottom\}$ are extracted from a forward pass with query $e$. **Step 2:** Top-2 adjectives are combined into contrastive descriptor $o^*$ with flip mapping $\phi$. **Step 3:** Residual difference $\Delta \mathbf{r}$ between target pass $e$ and contrastive pass $\phi(o^*)$ isolates the spatial grounding signal. Self-contradictory descriptors (e.g., "right-left") are replaced with "center".

where $\Delta \mathbf{a}_{\ell,h}$ reveals the reweighting of patch value vectors that produces the differential output: patches with large positive $\Delta \mathbf{a}_p$ are those the head actively selected to write spatial information distinguishing target from contrastive descriptions. This complements the $s_{\ell,h,p}^{\mathrm{IP}}$ score by measuring which patches were actually accessed by the attention mechanism.

## 4.5   Query Difference Score

Queries determine which key features are extracted: by steering queries toward discriminative spatial features, we isolate the specific patch representations that drive correct output generation, revealing the retrieval mechanism underlying spatial grounding. We compute the query difference for each layer $\ell$ and attention head $h$:

$$\Delta \mathbf{Q}_{\ell,h} = \mathbf{Q}_{\ell,h}^{(A)} - \mathbf{Q}_{\ell,h}^{(B)} \in \mathbb{R}^{d_h} \tag{8}$$

This difference vector isolates the features that differentiate the target region from the contrastive region in the model's query representation. We then compute the attention scores that this differential query produces with the image keys:

$$\mathbf{s}_{\ell,h}^{\mathrm{Q\text{-}diff}} = \Delta \mathbf{Q}_{\ell,h} \mathbf{K}_{\ell,h} [\mathrm{img}_{\mathrm{start}} : \mathrm{img}_{\mathrm{end}}]^{\top} \in \mathbb{R}^{P^2} \tag{9}$$

where $\mathbf{K}_{\ell,h}[\mathrm{img}_{\mathrm{start}} : \mathrm{img}_{\mathrm{end}}]$ extracts the key vectors for the $24 \times 24$ image patch tokens. Critically, since $\mathbf{K}_{\ell,h}$ is identical across both passes (same image), computing $\Delta \mathbf{Q}_{\ell,h} \mathbf{K}_{\ell,h}^{\top}$ measures which patches would be retrieved differently when the query changes from contrastive to target description. This isolates the spatial signal without confounding from different visual inputs. Positive scores indicate patches aligned with the target description, while negative scores indicate alignment with the contrastive region.

## 4.6   Attention Head Importance Ranking for Visual Grounding

We now combine these computed complementary scores to identify the most important attention heads for visual grounding. We rank each head by computing the weighted sum:

$$r_{\ell,h} = \sum_{p=1}^{P^2} \Delta \mathbf{a}_{\ell,h}[p] \cdot s_{\ell,h,p}^{\mathrm{IP}} \tag{10}$$

This criterion identifies heads that simultaneously satisfy two conditions: **1.** Isolates the features related to the target object ($\Delta \mathbf{a}_{\ell,h}[p] > 0$), **2.** Extract value information aligned with the differential representation ($s_{\ell,h,p}^{\mathrm{IP}} > 0$), marking these heads as components of the spatial localization circuit.

The resulting ranking matrix $\mathbf{R} \in \mathbb{R}^{N \times H}$ quantifies each head's contribution to differential localization, where $N$ is the number of layers and $H$ the number of heads per layer. We select the top-$k$ heads from $\mathbf{R}$ for targeted intervention in the localization process.

### 4.6.1 Top-$k$ Head Selection and Heatmap Aggregation

We select the top-$k$ attention heads with the highest importance rankings:

$$\mathcal{T} = \{(\ell_i, h_i)\}_{i=1}^k \text{ where } (\ell_i, h_i) = \underset{(\ell,h)\notin\mathcal{T}}{\arg\max} \, r_{\ell,h} \tag{11}$$

For each selected head $(\ell_i, h_i)$, we extract its query difference scores and apply ReLU to focus on positive contributions:

$$\tilde{\mathbf{s}}_i = \text{ReLU}(\mathbf{s}_{\ell_i,h_i}^{\text{Q-diff}}) \in \mathbb{R}^{P^2} \tag{12}$$

We aggregate these query difference scores using importance-weighted averaging:

$$\mathbf{H}_{\text{flat}} = \frac{\sum_{i=1}^k r_{\ell_i,h_i} \cdot \tilde{\mathbf{s}}_i}{\sum_{p=1}^{P^2} \left( \sum_{i=1}^k r_{\ell_i,h_i} \cdot \tilde{\mathbf{s}}_i[p] \right)} \tag{13}$$

The flattened heatmap is reshaped to a 2D spatial grid. The resulting heatmap combines information from the most important attention heads for localizing the target object, highlighting the image patches that contain the target object most strongly.

### 4.6.2 Heatmap Smoothing and Bounding Box Refinement

To reduce noise while preserving spatial structure, we apply multi-scale Gaussian smoothing with two different kernel sizes (sigma values of 0.5 and 1.5) and blend them with weights 0.7 and 0.3 respectively. We then threshold the smoothed heatmap at 70% of its maximum value to create a binary mask, apply morphological hole filling to remove small gaps, and extract the largest connected component as the final segmentation mask.

### 4.6.3 Fusion with Open-Vocabulary Detector:

To refine spatial precision, we leverage a pre-trained open-vocabulary detector Cheng et al. (2024) to generate a set of object proposals from the input image. The final bounding box is determined by computing IoU scores between our heatmap-derived candidate and all detector proposals, selecting the proposal with maximum overlap. This fusion combines the heatmap's semantic cues with the detector's localization precision, producing tighter and more accurate bounding boxes.

## 5 Experiments

### 5.1 Experimental Setup

**Models.** To demonstrate the generalizability of our method, we conduct experiments using four different LVLMs: LLaVA-1.5-7B Liu et al. (2023), LLaVA-1.5-13B Liu et al. (2023), BakLLaVA-v1 Liu et al. (2024a), and Qwen2-VL-7B-Instruct Wang et al. (2024a), which employs a distinct vision encoder and dynamic resolution processing mechanism, fundamentally different from the LLaVA-style linear projection architecture. We evaluate different configurations of localization heads across all model variants.

**Benchmarks.** We evaluate on four benchmarks: RefCOCO, RefCOCO+ Kazemzadeh et al. (2014), RefCOCOg Hu et al. (2016), and Flickr30K Entities Plummer et al. (2015). RefCOCO and RefCOCO+ include validation, testA (person-focused), and testB (object-focused) splits. RefCOCO emphasizes spatial descriptions, while RefCOCO+ focuses on appearance attributes. RefCOCOg contains longer, more detailed

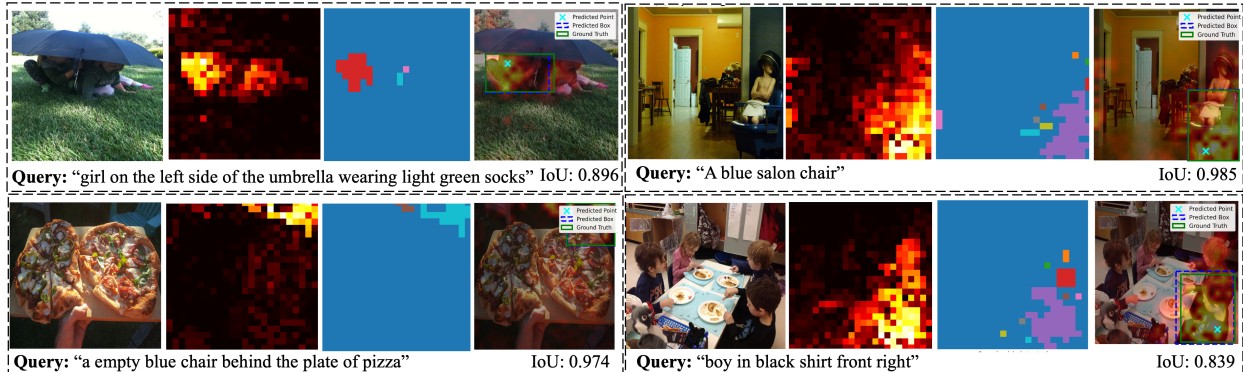

Figure 4: Localization results using contrastive prompting for head selection in LLaVA-1.5-7B Liu et al. (2023), each box displaying (from left) the input image, computed attention heatmap, connected component, and final prediction on challenging cases involving visual ambiguity, low-saliency targets, and intricate spatial arrangements.

expressions than both. Flickr30K Entities provides a distinct image domain and annotations across five entity categories (people, clothing, animals, vehicles, and instruments/other) relative to the COCO-derived benchmarks, allowing us to assess generalization. All ablation studies are performed on the RefCOCO validation set.

**Baselines.** We benchmark our approach against three categories of methods: (1) fine-tuning approaches, encompassing specialized visual grounding training based SOTA model UNINEXT Lin et al. (2023); LVLMs adapted by SOTA models CogVLM Wang et al. (2024b), MDETR Kamath et al. (2021), GLIP-L Li et al. (2022), and without LVLMs adapted by ZSGNet Sadhu et al. (2019), BAN Kim et al. (2018) for localization tasks, (2) training-free approach such as CPT-Blk Yao et al. (2024), ReCLIP Subramanian et al. (2022), Han et al. (2024), and GroundVLP Shen et al. (2024). Note that GroundVLP requires mapping predictions to dataset vocabulary categories, whereas our method performs purely spatial localization without category supervision, (3) head-selection based visual grounding method Kang et al. (2025), which selects attention heads using spatial-entropy statistics computed over 1,000 labeled RefCOCO samples. Unlike all baselines, our method requires no task-specific training, labeled grounding data, or category vocabulary.

## 5.2 Main Results

Table 2 shows that our training-free LVLM-based model outperforms existing training-free methods on RefCOCO, while achieving competitive performance on RefCOCO+/g. Table 3 shows that proposed method outperforms on Flickr30k dataset as well. We define Intersection over Union (IoU) as the area of overlap between the predicted and ground-truth bounding boxes divided by the area of their union, and mean IoU (mIoU) as the IoU averaged over all samples. We evaluate performance using two metrics: (1) Acc@0.5, the percentage of predictions with IoU $\geq$ 0.5, and (2) point-in-box accuracy (point acc.), the percentage of predictions where the predicted center point falls within the ground truth bounding box. Importantly, Acc@0.5 results incorporate open-vocabulary detector fusion for bounding box boundary refinement, while point-in-box accuracy is reported without any detector, reflecting the pure contribution of our contrastive attention-head localization framework.

We observe a trade-off between precise boundary localization and semantic language understanding. LVLMs semantically understand what and where the target object, but not the exact boundaries as well as specialized models do. Consequently, point accuracy substantially outperforms bounding box accuracy (Acc@0.5), which necessitates accurate boundary prediction.

Table 2: Comparison of our method against fine-tuning based and training-free baselines on visual grounding benchmarks. Fine-tuned models utilize the training data from each dataset. Red values indicate highest point localization within the object boundary (in %). Blue values indicate highest Acc@0.5 (in %). † denotes methods requiring labeled grounding samples for head selection.

| Method | Venue | RefCOCO | | | RefCOCO+ | | | RefCOCOg | |
|---|---|---|---|---|---|---|---|---|---|
| | | val | testA | testB | val | testA | testB | val | test |
| *Fine-tuning based methods* | | | | | | | | | |
| UNINEXT Lin et al. (2023) | CVPR 2023 | 91.43 | 93.73 | 88.93 | 83.09 | 87.90 | 76.15 | 86.91 | 87.48 |
| *Fine-tuning based methods w/ LVLMs* | | | | | | | | | |
| CogVLM-17B Wang et al. (2024b) | NeurIPS 2024 | 92.76 | 94.75 | 88.99 | 88.68 | 92.91 | 83.39 | 89.75 | 90.79 |
| *Training-free methods (no LVLM, no labeled samples)* | | | | | | | | | |
| CPT-Blk Yao et al. (2024) | AI Open 2024 | 26.90 | 27.50 | 27.40 | 25.40 | 25.00 | 27.00 | 32.10 | 32.30 |
| ReCLIP Subramanian et al. (2022) | ACL 2022 | 45.78 | 46.10 | 47.07 | 47.87 | 50.10 | 45.10 | 59.33 | 59.01 |
| Han et al. Han et al. (2024) | CVPR 2024 | 48.24 | 48.40 | 49.15 | 45.64 | 47.59 | 42.79 | 57.60 | 56.64 |
| *Training-free method w/ LVLM (ground-truth category provided)* | | | | | | | | | |
| GroundVLP Shen et al. (2024) | AAAI 2024 | 65.00 | 73.50 | 55.00 | 68.80 | 78.10 | 57.30 | 74.70 | 75.00 |
| *Few-shot head selection w/ LVLMs (†1,000 labeled samples required for head selection)* | | | | | | | | | |
| Kang et al.† Kang et al. (2025) | CVPR 2025 | 87.20 | 90.00 | 83.30 | 82.70 | 88.50 | 74.00 | 84.30 | 85.50 |
| *Zero-shot automatic head discovery w/ LVLMs (no labeled samples, no ground-truth category)* ***(proposed method)*** | | | | | | | | | |
| LLaVA-1.5-7B | – | 70.17 | 76.35 | 63.04 | 59.11 | 69.52 | 47.27 | 63.38 | 63.07 |
| LLaVA-1.5-13B | – | 69.35 | 76.35 | 62.71 | 59.45 | 68.46 | 50.06 | 64.74 | 65.04 |
| BakLLaVA-v1 | – | 66.96 | 74.03 | 56.57 | 56.45 | 67.44 | 45.27 | 59.26 | 59.75 |
| Qwen2-VL-7B | – | 63.57 | 68.40 | 59.60 | 57.59 | 59.28 | 47.40 | 54.20 | 52.40 |
| LLaVA-1.5-7B (Point Acc.) | – | 79.49 | 84.76 | 74.42 | 67.70 | 77.57 | 58.95 | 71.72 | 71.53 |
| LLaVA-1.5-13B (Point Acc.) | – | 78.82 | 83.80 | 73.59 | 68.15 | 76.91 | 59.07 | 71.81 | 73.64 |
| BakLLaVA-v1 (Point Acc.) | – | 74.91 | 81.67 | 69.94 | 63.39 | 73.27 | 53.56 | 66.43 | 65.50 |
| Qwen2-VL-7B (Point Acc.) | – | 73.40 | 76.80 | 71.20 | 59.03 | 64.61 | 54.20 | 62.80 | 59.80 |

Figure 4 depicts the step-by-step localization process. Each example shows four sequential stages: the original image with its textual query, the computed attention heatmap after applying multi-scale Gaussian smoothing, the dominant connected region obtained through morphological hole filling, and the final bounding box output. Our method achieves reliable localization across complex scenarios.

## 5.3 Ablation Studies

**Number of Localization Heads.** We perform ablation experiments to validate our head selection design. We investigate the effect of varying $(k)$ on visual grounding performance. Table 4 presents the results of our training-free framework with different $(k)$ values across multiple LVLMs. We observe that performance consistently improves as k increases from 1 to 10, indicating that top-ranked heads complement each other to provide more accurate localization of referred objects. Using only a few heads (k=1 or 3) yields suboptimal results, suggesting that individual heads capture only partial spatial information and multiple heads are necessary to gather complete grounding signals. However, the performance gains plateau around $(k = 10)$, with further increases providing diminishing returns. These results indicate that spatial grounding in LVLMs is distributed across roughly 5–10 attention heads, and no single head captures the full localization signal. This supports our hypothesis that spatial reasoning is encoded through complementary information spread across multiple discriminative attention mechanisms.

We investigate the relationship between attention head selection and grounding performance in Fig. 5. The negative Spearman correlation between selection rank and average IoU — $\rho = -0.40$ (LLaVA-1.5-7B), $\rho = -0.18$ (LLaVA-1.5-13B), $\rho = -0.01$ (BakLLaVA-v1), $\rho = -0.18$ (Qwen2-VL-7B) — confirms that our scoring criterion is meaningful: heads ranked higher by our contrastive scoring consistently achieve better localization, rather than selection being arbitrary. The strength of this correlation varies across architectures,

Table 3: Comparison on Flickr30k Entities benchmark (in %). R@1 denotes Recall@1 with IoU $\geq$ 0.5 threshold.

| Method | Venue | Val (R@1) | Test (R@1) |
|---|---|---|---|
| *Supervised methods with Vision-Language Pre-training* | | | |
| MDETR Kamath et al. (2021) | ICCV 2021 | 83.6 | 84.3 |
| GLIP-L Li et al. (2022) | CVPR 2022 | 86.7 | 87.1 |
| *Supervised methods without Vision-Language Pre-training* | | | |
| ZSGNet Sadhu et al. (2019) | ICCV 2019 | — | 63.39 |
| BAN Kim et al. (2018) | NeurIPS 2018 | — | 69.69 |
| *Training-free methods* | | | |
| CPT-Blk Yao et al. (2024) | AI Open 2024 | 27.06 | 27.57 |
| GroundVLP Shen et al. (2024) | AAAI 2024 | 63.76 | 63.89 |
| *Zero-shot automatic head discovery w/ LVLMs **(proposed method)*** | | | |
| LLaVA-1.5-7B | — | 64.23 | 64.01 |
| LLaVA-1.5-13B | — | 63.39 | 52.58 |
| BakLLaVA-v1 | — | 59.40 | 50.41 |
| Qwen2-VL-7B | — | 48.28 | 50.17 |
| LLaVA-1.5-7B (Point Acc.) | — | 78.02 | 73.15 |
| LLaVA-1.5-13B (Point Acc.) | — | 68.70 | 73.00 |
| BakLLaVA-v1 (Point Acc.) | — | 63.75 | 65.37 |
| Qwen2-VL-7B (Point Acc.) | — | 46.94 | 39.85 |

Table 4: Performance variation with respect to the number of localization attention heads ($k$) on the Ref-COCO validation set (in %).

| Method | Number of Localization Heads ($k$) | | | | |
|---|---|---|---|---|---|
| | 1 | 3 | 5 | 10 | 20 |
| LLaVA-1.5-7B | 64.6 | 70.4 | 71.8 | **72.6** | 71.8 |
| LLaVA-1.5-13B | 55.2 | 64.6 | 68.0 | 71.0 | **71.4** |
| BakLLaVA-v1 | 56.4 | 64.0 | 67.2 | **68.8** | 68.2 |
| Qwen2-VL-7B | 62.4 | 69.8 | 71.0 | **72.5** | 71.5 |
| Average | 59.6 | 67.2 | 69.5 | **71.2** | 70.7 |

with LLaVA-1.5-7B showing the clearest alignment between score rank and localization quality, suggesting that spatial grounding is more concentrated in specialized heads in smaller models.

**LVLM Layer Analysis for Attention Head Selection.** Table 5 reports the spatial distribution of discovered attention heads across network depth for all four models. LLaVA-style models concentrate spatial grounding in middle layers: LLaVA-1.5-7B selects 79.5% of heads from middle layers (10–20), with $\mathcal{L}_{14}$ serving as the dominant spatial grounding hub (246.8% selection frequency). LLaVA-1.5-13B exhibits more distributed pattern, with $\mathcal{L}_{21}$ as its dominant layer (158.2%). This suggests that spatial grounding emerges at a consistent relative depth rather than a fixed absolute layer index, scaling with model size.

In contrast, BakLLaVA-v1 and Qwen2-VL-7B rely predominantly on later layers: BakLLaVA-v1 concentrates 73.8% of selections in late layers (16–23) with $\mathcal{L}_{24}$ dominant (233.1%), while Qwen2-VL-7B distributes selections broadly across middle and late layers with a notably lower peak frequency ($\mathcal{L}_{22}$: 75.5%), reflecting its distinct dynamic-resolution architecture which processes visual tokens differently from LLaVA-style linear projection. This late-layer reliance coincides with weaker Spearman correlations ($\rho = -0.01$ for BakLLaVA-v1, $\rho = -0.18$ for Qwen2-VL-7B) and lower overall localization performance, suggesting that middle-layer grounding is more effective for spatial localization than late-layer grounding. The near-absence of early-layer

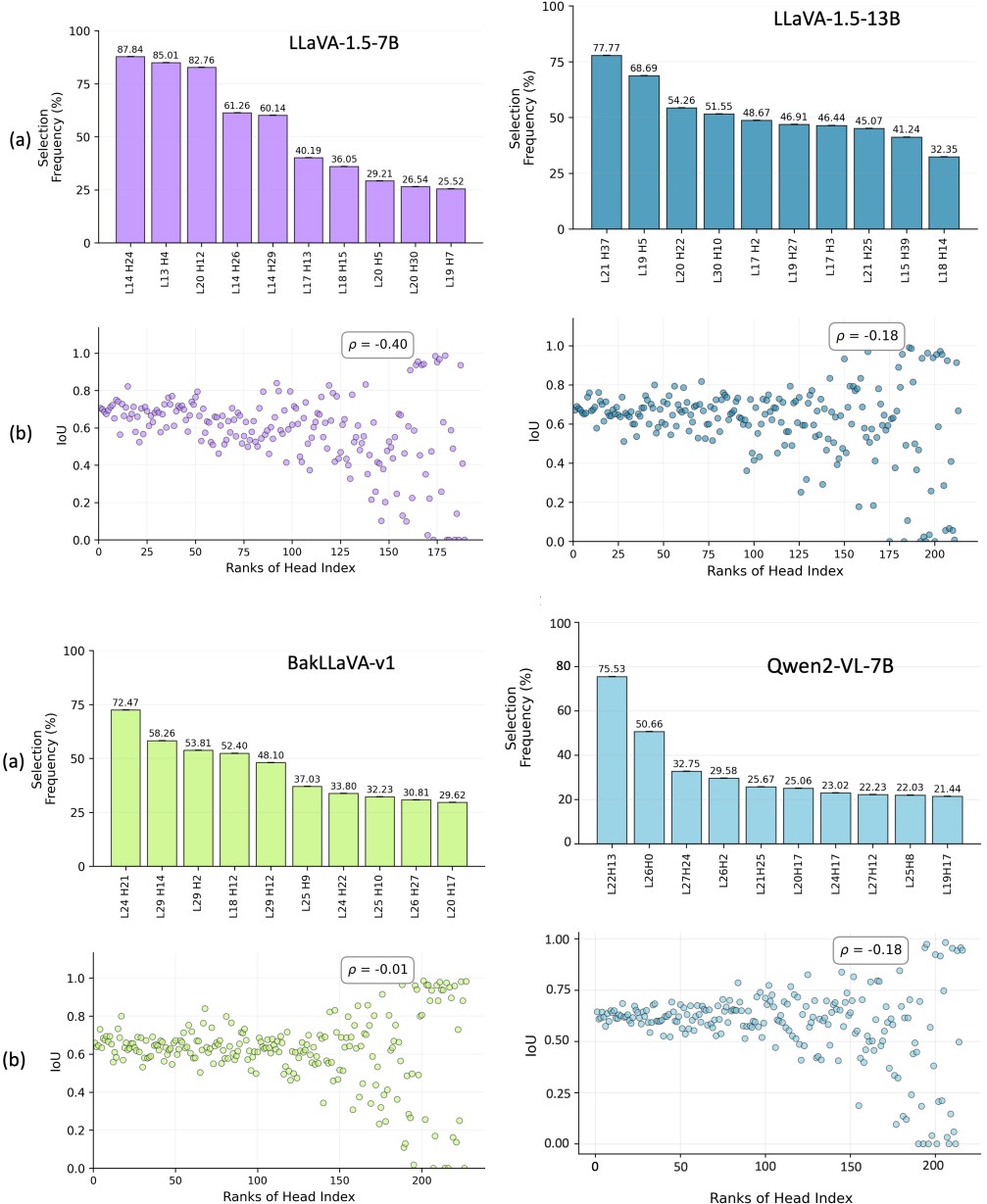

Figure 5: Analysis of attention head selection for visual grounding on LVLMs across RefCOCO validation set. (a) Selection frequency showing the percentage of samples where each of the top-10 most frequently selected heads was chosen. (b) Correlation between head selection rank and average IoU performance, with Spearman's $\rho$ indicating the relationship between selection frequency and localization accuracy.

selections across all models suggests that grounding mechanisms emerge only after sufficient cross-modal feature abstraction has occurred.

**Component Analysis.** Analysis of the top-10 discovered attention heads reveals two complementary localization strategies, as shown in Table 6. Heads such as $\mathcal{L}_{14}\mathcal{H}_{24}$ ($S^{\text{Q-diff}} = -424$, mIoU=0.756) operate through *distractor suppression* — large negative query difference scores indicate the head attends away from the contrastive region, filtering background patches. Conversely, heads such as $\mathcal{L}_{13}\mathcal{H}_4$ ($S^{\text{Q-diff}} = +401$, mIoU=0.772) operate through *target activation* — strong positive scores reflect direct alignment with the

Table 5: Spatial layer distribution of discovered attention heads in LVLMs. Selections denotes cumulative head selection count across all RefCOCO validation set with $k = 10$. Dominant layer denotes the single layer with highest selection frequency.

| Model | Layer Position | Avg IoU | Selection | Dominant Layer (Freq.) |
|---|---|---|---|---|
| LLaVA-1.5-7B | Early (0-9) | 0.0 | 0 | |
| | Middle (10-20) | 68.6 | 70,001 | $\mathcal{L}_{14}$ (246.8%) |
| | Late (21-26) | 66.2 | 18,109 | |
| LLaVA-1.5-13B | Early (0-12) | 70.1 | 499 | |
| | Middle (13-26) | 67.7 | 60,073 | $\mathcal{L}_{21}$ (158.2%) |
| | Late (27-39) | 66.3 | 27,538 | |
| BakLLaVA-v1 | Early (0-7) | 0.0 | 0 | |
| | Middle (8-15) | 69.9 | 23,097 | $\mathcal{L}_{24}$ (233.1%) |
| | Late (16-23) | 63.5 | 65,013 | |
| Qwen2-VL-7B | Early (0-13) | 63.3 | 529 | |
| | Middle (14-26) | 59.7 | 41,005 | $\mathcal{L}_{22}$ (75.5%) |
| | Late (27-39) | 62.0 | 46,576 | |

Note: Dominant Layer frequency represents the cumulative sum of selection rates across all heads within that layer, and may exceed 100% when multiple heads are each selected with high frequency.

Table 6: Top-10 attention heads ranked by combined scoring metric using LLaVA-1.5-7B. Heads exhibit two complementary localization strategies: *target activation* (positive $S^{\text{Q-diff}}$, e.g., L13H4) and *distractor suppression* (negative $S^{\text{Q-diff}}$, e.g., L14H24, L14H29), both achieving competitive mIoU through distinct mechanisms.

| Rank | Layer | Head | $S^{\text{IP}}$ | $\Delta\mathbf{a}$ | $S^{\text{Q-diff}}$ | mIoU | Point Acc. |
|---|---|---|---|---|---|---|---|
| 1 | 14 | 24 | -50 | +0.21 | -424 | **0.756** | **0.877** |
| 2 | 20 | 12 | +53 | -0.24 | -219 | 0.687 | 0.766 |
| 3 | 13 | 4 | +48 | +0.22 | +401 | **0.772** | **0.877** |
| 4 | 18 | 15 | +81 | -0.17 | +202 | 0.658 | 0.826 |
| 5 | 14 | 26 | +27 | +0.19 | +161 | 0.644 | 0.788 |
| 6 | 14 | 29 | +8 | -0.05 | -868 | 0.717 | 0.826 |
| 7 | 17 | 13 | -1 | -0.10 | -35 | 0.747 | **0.867** |
| 8 | 26 | 21 | +159 | +0.05 | +548 | 0.666 | 0.781 |
| 9 | 19 | 7 | +145 | +0.00 | +111 | 0.466 | 0.677 |
| 10 | 20 | 5 | +27 | -0.17 | -99 | 0.566 | 0.691 |

target region's key vectors. Both strategies achieve comparable point accuracy (0.877), confirming that suppression and activation represent equally viable localization mechanisms rather than one being strictly superior.

This coexistence explains the near-zero $S^{\text{Q-diff}}$ correlation in Table 7 ($r = -0.071$): suppression and activation heads carry opposing signs but both contribute positively to localization, canceling in a linear measure. By contrast, attention difference $\Delta\mathbf{a}$ ($r = +0.156$) and patch attribution $S^{\text{IP}}$ ($r = +0.148$) positively correlate with IoU, confirming that *where* the model attends and *what* information is propagated are stronger localization indicators than query-key alignment strength alone. High variance in component scores across samples (e.g., $\mathcal{L}_{14}\mathcal{H}_{24}$: 0.67±0.56) further indicates adaptive activation — heads modulate their contribution based on expression complexity and visual context while maintaining consistent localization performance. This motivates our importance-weighted aggregation (Eq. 13), which combines both strategies rather than filtering by sign. Table 8 demonstrates that contrastive head selection effectively isolates object-relevant patterns, substantially outperforming raw attention (79.49% vs 22.52% point accuracy).

Table 7: Pearson correlation coefficients between component scores and localization performance (IoU) across all heads. The near-zero $S^{\text{Q-diff}}$ correlation reflects a bimodal head strategy — suppression heads (e.g., L14H24) and activation heads (e.g., L13H4) carry opposing signs but both achieve high localization, canceling in a linear measure (see Table 6).

| Component | Correlation with IoU |
|---|---|
| Attention Difference Score ($\Delta\mathbf{a}$) | +0.156 |
| Patch Attribution Score ($S^{\text{IP}}$) | +0.148 |
| Query Difference Score ($S^{\text{Q-diff}}$) | -0.071 |
| Combined Ranking | +0.097 |

Table 8: Comparison of raw attention and contrastive attention (in %).

| Method | Point Acc. | mIoU | Acc@0.5 |
|---|---|---|---|
| Raw Attention | 22.52 | 27.95 | 26.59 |
| Contrastive | 79.49 ↑ | 68.11 ↑ | 70.17 ↑ |

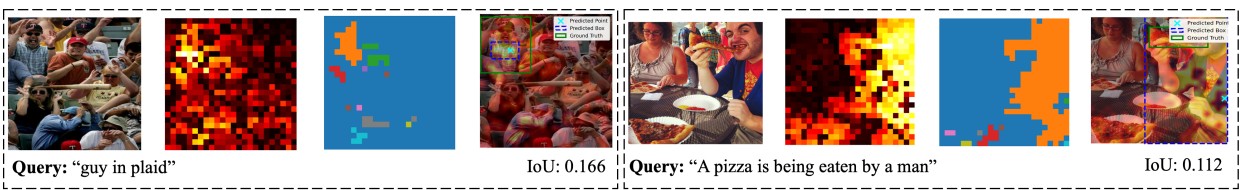

Figure 6: Failure case analysis of our contrastive prompting approach with LLaVA-1.5-7B.

**Query Type Analysis.** Table 9 presents localization performance across five referring expression categories. Attribute-based queries yield the strongest overall performance (Point Acc.: 84.4%, Acc@0.5: 76.9%), confirming that visual properties such as color and size provide reliable contrastive cues for spatial descriptor extraction. Action queries achieve competitive results (Point Acc.: 90.9%, Acc@0.5: 77.3%). Relational queries prove most challenging (Point Acc.: 75.7%, Acc@0.5: 62.7%), suggesting that object-object spatial relations are harder to ground through attention alone — likely because relational expressions provide weaker directional contrast for contrastive prompt generation. Complex queries, despite being the largest category ($n = 3{,}327$), achieve competitive performance across all metrics (Point Acc.: 76.4%, mIoU: 64.8%, Acc@0.5: 66.7%), demonstrating that our framework handles compositional expressions robustly. Positional queries perform consistently across all metrics (Point Acc.: 78.2%, mIoU: 67.6%, Acc@0.5: 69.1%), as expected given that spatial adjectives are directly leveraged by our contrastive descriptor mechanism.

## 5.4 Understanding LVLMs When They Fail

We analyze failure cases to understand the mechanistic boundaries of our approach. As shown in Figure 6, our method reveals two key properties of vision-language attention: (1) Attention heads prioritize semantically salient regions (e.g., faces for "person") over complete object extents. (2) Semantic ambiguity produces diffuse or multi-modal attention patterns. These cases show that our approach faithfully surfaces *where the model attends* during spatial grounding, which may differ from ground-truth object boundaries.

## 5.5 Application of Proposed Approach in Segmentation

The effectiveness of our contrastive method stems from how localization heads naturally group semantically similar patches under shared object features—all patches corresponding to "car" tend to activate similar feature representations. By isolating the query's contribution through contrastive attention, we suppress background activations (heatmap weight on patches outside the target object) while preserving spatially-

Table 9: Localization performance breakdown by referring expression. Values are reported per category (in %). $n$ denotes sample count per category.

| Query Type | Sample Expression | $n$ | Point Acc. | mIoU | Acc@0.5 |
|---|---|---|---|---|---|
| Position | "left person", "bottom chair" | 2,415 | 78.2 | 67.6 | 69.1 |
| Attribute | "red car", "small dog" | 854 | 84.4 | 73.1 | 76.9 |
| Relation | "person on horse", "cup on table" | 375 | 75.7 | 62.5 | 62.7 |
| Action | "person walking" | 88 | 90.9 | 75.9 | 77.3 |
| Complex | "large red car on the left" | 3,327 | 76.4 | 64.8 | 66.7 |

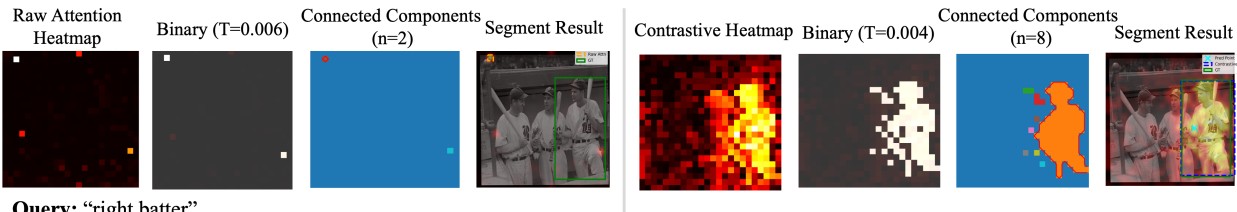

**Query:** "right batter"

Figure 7: Comparison of raw attention (left) vs. contrastive heatmap (right) segmentation pipelines. Contrastive attention suppresses background activations and produces spatially-coherent, mask-like patterns that align closely with target object boundaries

coherent, mask-like patterns centered on target objects. Unlike approaches that expand vocabulary with specialized tokens Lai et al. (2024) or identify object-informative heads via training set Kang et al. (2025), our method directly exploits this inherent grouping behavior without additional training or architectural modifications. As Fig. 7 demonstrates, the resulting masks align closely with ground-truth boundaries. One limitation is that semantically-related but distinct object aspects may also be captured.

### 5.6 Broader Impact

Our method improves training-free visual grounding in frozen LVLMs, benefiting assistive vision, image retrieval, robotics, and interpretability. Like all grounding technology it is dual-use — it could be repurposed for surveillance or tracking, and its training-free nature lowers the barrier to such deployment. Two factors limit this risk: the method surfaces representations the frozen LVLM and off-the-shelf detectors already encode rather than creating new capability, and its accuracy remains well below the fine-tuned systems high-stakes deployments would realistically use. Downstream use must comply with applicable privacy regulations; we view making LVLM grounding transparent and auditable as a net contribution to responsible AI.

## 6 Conclusion

In this paper, We propose a training-free attention-head discovery framework that identifies discriminative localization heads for visual grounding without labeled samples, or architectural modifications. We introduce four complementary scores, e.g., the attribution score via output projection, image patch attribution score, attention difference score, and query difference scores to rank attention heads. By aggregating signals from only the top-10 heads out of thousands of attention heads, we are able to achieve improved performance on RefCOCO/+/g and Flickr30k benchmarks. Our framework automatically discovers the discriminative subset of attention heads that enable precise spatial localization without fine-tuning and architectural modifications. Our method produces precise spatial grounding that also aligns closely with target object shapes. **Future Work.** This observation opens a promising research direction toward leveraging contrastive attention patterns as supervision for zero-shot segmentation masks without additional training nor heuristics.

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
