# OpenReview forum: "Few Contrastive Attention Heads Enable Visual Grounding in Large Vision-Language Models"
_TMLR — Under review for TMLR_

### Review · Reviewer_F3ZS · 2026-05-28

**Summary Of Contributions:**

The paper studies the attention heads that give the major contribution to visual grounding tasks. In particular, the paper proposes a contrastive descriptor generation mechanism which generates two contrastive description sentences given a language-based input. Then, it introduces four scores (attribution via output projection, image patch attribution, attention difference and query difference scores) to help identify the attention heads that contribute most to the task. Such method is training-free. Finally, numerous experiments on visual grounding are performed to analyze the method and compares it to several existing methods.

**Additional Comments:**

I am not very familiar with this area.

**Audience:**

Yes

**Audience Explanation:**

Visual grounding is a large area in machine learning and has many applications. Thus, understanding its mechanism and improving the performance are essential.

**Broader Impact Concerns:**

N.A.

**Claims And Evidence:**

Yes

**Claims Explanation:**

The experimental results are discussed in detail and the accuracy seemed to be quite superior to the other methods mentioned in the paper. The introduction of the method is generally clear but can be made better. See "Requested Changes" below.

**Requested Changes:**

1. Page 4: For $Z_t$ it is better to use "t" as an indication for "text" instead of number of texts, so that it would not be confused with $Z_0$. And is it $V_{l,h} \in \mathbb{R}^{(P^2 + L) \times d_h}$ instead of $V_{l,h} \in \mathbb{R}^{(P^2 + t) \times d_h}$? If not please explain the meaning of $t$ here.

2. Page 5: In Section 4.1, what does "final token logits" refer to -- how close to the end of the token do the authors view as "final token logits"?

3. Page 5: In Section 4.1, please explain the meaning of $e, a_1, a_2, s_1, s_2$.Third, it would great if the authors could explain how and why they use (specifically) the flip map for generating the contrastive description, especially if "e" does not indicate the position of the object -- or must "e" indicate the position of the object?

4. Page 6: What is $v_p^{(l,h)}$?

5. Page 8, 12, 13: In Section 5, please explain the meaning of "IoU" and "mIoU".

6. Page 10: In Section 5.3, why does including 10 more heads (slightly) decreases the performance? Is it because more heads would obscure the accurate information? If so, it might be good to include a column of a large number of heads (but not all heads) for comparison. Moreover, how is the performance around 10, say, between 8 - 12?

7. Page 10: In Section 5.3, does the number of heads depend on the size of the models used (provided that it has more than the number of heads in the table)?

8. Page 14: In Section 5.5, please explain the meaning of "background activation".

---

> ### Author Response · Authors · 2026-07-14
> **Response to Reviewer F3ZS (1)**
>
> We thank the reviewer for their feedback and provide our response below; all corresponding clarifications and notation fixes have been incorporated into the revised manuscript.
>
> **Q1. Notation of $Z_t$ and the dimension of $V_{l,h}$**
>
> *On the subscript of $Z_t$.* It was intended only as a modality marker for "text" (paralleling $v$ for vision in $Z_v$), not as a token count or layer index. Since the same letter was also used for the token count, we have revised the notation so the number of text tokens is consistently denoted $L$, and the subscript $t$ is reserved purely for the text modality. With the revised convention, the query, key, and value matrices all share the dimension $(P^2+L)\times d_h$, consistent with the input $Z_0=[Z_v;Z_t]\in\mathbb{R}^{(P^2+L)\times d}$.
>
> **Q2. Meaning of "final token logits" in Section 4.1**
>
> By "final token logits" we mean the next-token logit distribution at the last token position of the ASSISTANT prompt, i.e., the logits the model would use to predict the first generated token. Following the information-flow argument (the last token aggregates the preceding context under autoregressive decoding), we use this single position's logits to rank spatial adjectives and nouns.
>
> **Q3. Meaning of $e$, $a_1$, $a_2$, $s_1$, $s_2$, and the rationale for the flip map**
>
> *Notation.* $e$ is the referring expression (the target query). $\mathcal{A}=\{\text{left, right, top, bottom}\}$ is the spatial-adjective vocabulary; $a_1$ and $a_2$ are the top-1 and top-2 adjectives in $\mathcal{A}$ ranked by their logits at the final prompt position, with scores $s_1, s_2$. If $s_1-s_2<1$ (near-tie), we form a compound descriptor $o^*=a_1$-$a_2$ (e.g., "top-left"); otherwise $o^*=a_1$.
>
> *On the flip map.* $e$ is **not** required to contain a positional term. The adjectives $a_1, a_2$ are not parsed from $e$; they are read from the model's own logit distribution when prompted to describe $e$. Even for a purely appearance-based expression such as "red car," the model assigns higher logit mass to the adjective matching where it localizes the object, so $o^*$ reflects the model's *inferred* position of $e$, not any explicit position word. The flip map $\phi$ (left $\leftrightarrow$ right, top $\leftrightarrow$ bottom) then directs the contrastive pass toward the opposite region while excluding $e$, so shared non-spatial semantics cancel in the residual difference and the target's spatial signal is isolated.
>
> **Q4. Definition of** $v_p^{(\ell,h)}$
>
> $v_p^{(\ell,h)} \in \mathbb{R}^{d_h}$ is the value vector of the $p$-th image-patch token at layer $\ell$, head $h$ — i.e., the row of the value matrix $V_{\ell,h}$ (Section 3.2) corresponding to image patch $p$, for $p \in \{1, \dots, P^2\}$. Equation 6 computes the alignment between this patch value vector and the normalized output-projection direction $\hat{z}_{\ell,h}$ to measure the magnitude of each patch's contribution.
>
> **Q5. Meaning of "IoU" and "mIoU"**
>
> IoU (Intersection-over-Union) is the standard overlap metric between the predicted and ground-truth bounding boxes: the area of their intersection divided by the area of their union (1 = perfect overlap, 0 = no overlap). mIoU (mean IoU) is the IoU averaged over all evaluation samples.
>
> **Q7. Does the number of heads depend on model size?**
>
> No---the number of localization heads is held fixed across model sizes and does not scale with a model's total head count. For every referring expression, the method discovers a fixed-size set of $k{=}10$ heads, regardless of whether the model has 32 heads per layer (LLaVA-1.5-7B, BakLLaVA-v1), 40 (LLaVA-1.5-13B), or 28 (Qwen2-VL-7B).
>
> We ablate $k\in\{1,3,5,10,20\}$ (Table 3) across four LVLMs of differing size and architecture. Three of the four peak at $k{=}10$, and the fourth (LLaVA-1.5-13B) is essentially flat between $k{=}10$ and $k{=}20$ ($71.0$ vs. $71.4$); we therefore fix $k{=}10$ for all models rather than tuning it per model. Performance rises steeply through small $k$ and plateaus by $k{\approx}5$--$10$, and the plateau location does *not* scale with model size---the larger 13B model does not require or benefit from more heads than the 7B. This indicates the number of localization-relevant heads is a stable, small quantity across models rather than a model-dependent one.
>
> **Q8. Meaning of "background activation"**
>
> Our heatmaps are spatial probability distributions over the $24\times24$ image patches, where each patch receives an activation value (the aggregated, ReLU'd query-difference score from Eqs. 12--13). By "background activation" we mean non-zero heatmap values on patches outside the target object---spurious weight the model assigns to background regions even though they are not the referent. The contrastive subtraction cancels these shared, non-target responses, concentrating activation on the target and yielding cleaner, mask-like segmentations (Figure 7).

---

> ### Author Response · Authors · 2026-07-14
> **Response to Reviewer F3ZS (2)**
>
> **Q6. Why performance slightly degrades beyond $k{=}10$, and fine-grained results for $k=8$–$12$**
>
> **(1) Comparison with a large number of heads.** Table 3 of the paper already contains a large-$k$ data point: the $k{=}20$ column, which doubles the head budget relative to $k{=}10$. The degradation at $k{=}20$ is consistent across all four evaluated LVLMs --- LLaVA-1.5-7B ($72.6 \rightarrow 71.8$), BakLLaVA-v1 ($68.8 \rightarrow 68.2$), and Qwen2-VL-7B ($72.5 \rightarrow 71.5$), with LLaVA-1.5-13B essentially saturating ($71.0 \rightarrow 71.4$) --- an average point-accuracy drop from $71.2$ to $70.7$.
>
> The slight decrease arises because heads are ranked by our importance criterion (Eq. 13), so heads beyond the top ${\sim}10$ carry weaker, noisier localization signal. Since the heatmap is an importance-weighted average, including these lower-quality heads dilutes the sharp signal from the top heads and softens the peak---consistent with the negative rank--IoU correlation in Figure 5.
>
> **(2) Fine-grained analysis around $k{=}10$ ($k=8$--$12$).** Following the reviewer's suggestion, we performed a fine-grained sweep on the RefCOCO validation set with LLaVA-1.5-7B:
>
> | Metric | $k{=}8$ | $k{=}9$ | $k{=}10$ | $k{=}11$ | $k{=}12$ |
> |---|---|---|---|---|---|
> | Point Acc. | 79.33 | 79.53 | 79.70 | 79.75 | 79.69 |
> | mIoU | 67.83 | 67.89 | 67.95 | 68.04 | 68.08 |
> | Acc@0.5 | 69.77 | 69.88 | 69.95 | 70.09 | 69.93 |
>
> *Stability of the discovered heads.* The same five heads dominate selection at every $k$ in this range, with only minor re-ordering between ranks 2 and 3, and each head's average IoU remains essentially constant. This confirms the head-discovery mechanism is stable: varying $k$ around 10 only changes how many low-frequency auxiliary heads are appended to a fixed core circuit.
>
> Top-5 most frequently selected heads; each cell shows selection frequency in % (average IoU when selected):
>
> | Head | $k{=}8$ | $k{=}9$ | $k{=}10$ | $k{=}11$ | $k{=}12$ |
> |---|---|---|---|---|---|
> | L14H24 | 81.33 (0.712) | 83.63 (0.709) | 85.51 (0.708) | 87.23 (0.705) | 88.62 (0.705) |
> | L13H4 | 76.55 (0.703) | 80.13 (0.700) | 83.09 (0.698) | 85.69 (0.699) | 87.74 (0.700) |
> | L20H12 | 78.87 (0.683) | 81.13 (0.685) | 82.84 (0.685) | 84.13 (0.687) | 85.33 (0.688) |
> | L14H26 | 54.48 (0.674) | 59.04 (0.675) | 63.38 (0.678) | 66.76 (0.681) | 69.62 (0.684) |
> | L14H29 | 45.11 (0.685) | 51.46 (0.685) | 57.02 (0.691) | 61.98 (0.693) | 66.61 (0.695) |
>
> *Growth of the auxiliary head pool.* Increasing $k$ from 8 to 12 enlarges the pool of unique heads selected at least once from 179 to 207 (out of 1,024), i.e., each additional slot mostly recruits heads selected only occasionally and per-sample. The rank--IoU Spearman correlation weakens as $k$ grows ($\rho = -0.37 \rightarrow -0.27$), directly evidencing that the marginal heads admitted at larger $k$ are progressively less aligned with localization quality --- the mechanism underlying the eventual degradation at $k{=}20$ discussed in part (1).
>
> | | $k{=}8$ | $k{=}9$ | $k{=}10$ | $k{=}11$ | $k{=}12$ |
> |---|---|---|---|---|---|
> | Unique heads selected | 179 | 185 | 194 | 201 | 207 |
> | Spearman $\rho$ (rank vs. IoU) | $-0.371$ | $-0.394$ | $-0.372$ | $-0.303$ | $-0.275$ |
> | Top-1 head selection freq. (%) | 81.33 | 83.63 | 85.51 | 87.23 | 88.62 |
> | Top-10 cum. selection freq. | 4.64 | 4.98 | 5.28 | 5.57 | 5.81 |
>
> We have added these experiments and analysis to Appendix A.5 of the revised paper.

---

### Review · Reviewer_yFsZ · 2026-06-15

**Summary Of Contributions:**

This paper propose a solution to adapt LVLM for visual grounding (localize image regions related to NLP expressions) without requiring fine-tuning. The high level idea is to identify discriminative localization heads and then  aggregates signals from the topk head to form the heat map for image localization.

The way they find the localization attention heads is by inputing the model twice on target and contrastive text descriptions and calculate the residual.

**Additional Comments:**

After searching related paper, I find a highly relevant paper:

Your Large Vision-Language Model Only Needs A Few Attention Heads For Visual Grounding CVPR2025

This paper does not explicitly tell the contribution against this CVPR paper and per my evaluation the contribution is marginal.

**Audience:**

Yes

**Audience Explanation:**

This paper looks interesting at least to me.

**Claims And Evidence:**

No

**Claims Explanation:**

The proposed method cannot outperform Kang et al. The contribution of elinimating 1000 samples for head selection is marginal. Other procedures of this paper is a direct borrow of Kang et al. and the authors did not explicitly tell the connections of the two paper. I think this is a very unethical manner and I strongly suggest to reject such paper.

**Requested Changes:**

I personally am not working on image domain and is not very familar with the visual grounding task. This paper looks interesting to me but I would like to ask several questions to verifies the contribution of this paper.

1. Could you explain more on how we aggregates the heat map based on the identified attention heads. I am confused here by looking at Eq. (12) because it appears to me that an attention head should accept information from the full figure, and why using the output of the attention heads can help identify the important region in the figure?

2. Does exisitng paper use a similar way (see question 1) to extract the heat map with the attention heads? If so, is the contribution of this paper to identify the top-K heads with the proposed contrastive method?

I don't have much comments due to the unfaimilar with this field but at least it seems to me that the paper looks interesting.

---

> ### Author Response · Authors · 2026-07-15
> **Response to Reviewer yFsZ (1)**
>
> **The relationship to Kang et al. (2025) is stated explicitly in the submission**
>
> We respectfully but firmly push back on the characterization that we "did not explicitly tell the connections of the two papers." The submitted manuscript cites, describes, and empirically compares against Kang et al. (2025) in four separate places:
>
> 1. **Related Work (Sec. 2, final paragraph)** is devoted to Kang et al.: *"Kang et al. (2025) demonstrated that a small subset of attention heads in LVLMs is sufficient for visual grounding, identifying these heads by computing spatial-entropy statistics over 1,000 labeled RefCOCO samples. While effective, this approach couples head selection to a specific dataset distribution, requiring re-annotation for new domains. Our work addresses the same question --- which heads matter for grounding? --- but answers it without any labeled data."*
>
> 2. **Baselines (Sec. 5.1)** lists Kang et al. as a dedicated third baseline category: *"head-selection based visual grounding method Kang et al. (2025), which selects attention heads using spatial-entropy statistics computed over 1,000 labeled RefCOCO samples."*
>
> 3. **Table 1** reports Kang et al.'s numbers in their own labeled tier (*"Few-shot head selection w/ LVLMs († 1,000 labeled samples required for head selection)"*), visually separated from our zero-shot tier precisely so readers can see both the gap and the setting difference at a glance.
>
> 4. **Sec. 5.5** contrasts our mechanism with theirs (*"Unlike approaches that ... identify object-informative heads via training Kang et al. (2025) ..."*).
>
> We therefore ask the reviewer to reconsider the ethics assessment, which we believe is not supported by the text of the submission. We take the ethics concern very seriously and hope the above resolves it.

---

> ### Author Response · Authors · 2026-07-15
> **Response to Reviewer yFsZ (2)**
>
> **"Cannot outperform Kang et al." compares two different problems --- and Kang et al.'s own ablation shows their method cannot solve ours**
>
> The two methods answer the same scientific question under *different information budgets*, and the correct comparison must hold that budget fixed.
>
> Setting-level comparison — Kang et al. perform *offline, dataset-calibrated* head selection; we perform *per-sample, zero-shot* head discovery at inference time:
>
> | | Kang et al. (2025) | Ours |
> |---|---|---|
> | **Dataset samples for head selection** | 1,000 (RefCOCO train) | **0** |
> | **Head set** | fixed per model, dataset-coupled | discovered per (image, expression) |
> | **Selection signal** | attention sum + spatial entropy statistics | contrastive residual/attention/query analysis |
> | **Heatmap signal** | raw attention of fixed heads | ReLU'd query-difference scores, importance-weighted |
> | **New domain** | requires re-calibration on that distribution | no adaptation step of any kind |
> | **Causal evidence for selected heads** | --- | ablation + activation patching (**Appendix A.1, A.2**) |
>
> **Further distinctions from Kang et al.**
>
> 1. **The refinement step does not explain the comparison's outcome.** Complementing the matched-budget argument, our detector-only controls (evaluating the detector in isolation under expression-blind selection rules) show the detector alone reaches $\leq$34.6 Acc@0.5 --- so our 69.35 in the zero-label comparison is attributable to the head-discovery mechanism, not the box refinement.
>
> 2. **Causal validation, which no prior head-selection method provides.** Kang et al. validate their heads correlationally ($\rho > 0.7$); our head-ablation and activation-patching experiments show our discovered heads are *causally responsible* for grounding behavior. The criterion is causally motivated by construction: contrastive prompting induces two spatially opposed readings, and heads are ranked by their measured contribution to the output difference (Eq. 10). The same heads' query--key geometry yields our heatmaps (Eqs. 12--13), so head identification and heatmap extraction are two readouts of one circuit, not independent heuristics. Robustness also differs: their head-count ablation degrades sharply past its optimum (67.1 at $k{=}3$ → 58.9 at $k{=}5$, their Table 4), while ours is nearly flat from $k{=}10$ to $k{=}20$ (71.2 → 70.7, paper Table 4), suggesting contrastive scoring surfaces a deeper pool of usable heads.

---

> ### Author Response · Authors · 2026-07-15
> **Response to Reviewer yFsZ (3)**
>
> **Q1. How is the heatmap aggregated from the identified heads, given that head outputs receive information from the full image?**
>
> The key clarification is that the spatial signal is **not** the head's output token vector. In Eq. (12) we apply $\mathrm{ReLU}$ to the query-difference score $s^{Q\text{-diff}}_{\ell,h}\in\mathbb{R}^{P^2}$, which is itself a *per-patch* quantity. From Eq. (9),
> i.e. the difference between the *target* ($A$) and *contrastive* ($B$) query vectors, dotted against each image-patch key. Each of the $P^2$ components corresponds to exactly one image patch, so the score is spatial *by construction*.
>
> Because the two prompts differ only *after* the image tokens, causal masking guarantees the image-patch keys $K_{\ell,h}[\text{img}]$ are identical across both passes; hence $\Delta Q_{\ell,h}K_{\ell,h}^{\top}$ measures *which patches the head would attend to differently when the query shifts from the contrastive description to the target description*. The contrastive subtraction cancels patches that both passes attend to (shared background and non-spatial semantics) and isolates target-specific patches. Eq. (13) then combines these per-patch maps across the top-$k$ heads using the importance weights $r_{\ell_i,h_i}$.
>
> (Head *outputs* enter our method only through Eq. (10), where the projection $W_O^{(\ell)}[h]^{\top}\Delta\mathbf{r}$ is used to decide *which heads to trust* --- a per-head scalar --- never as the spatial map itself.)
>
> The reviewer's observation that outputs are spatially entangled is precisely why the heatmap is built from query--key scores instead; our new factorial ablation confirms this: swapping the aggregation signal to the attention-difference maps degrades the heatmap from a coherent object mask to scattered activations (mIoU 0.32--0.37 → 0.005--0.026; Acc@0.5 ≈ 0) and costs −25.3 points of pointing accuracy on average.
>
> Scoring-function ablation, pointing accuracy (%); the full combination is the best configuration in all 12 model × split settings:
>
> | Model | Split | Full | −$\Delta a$ (rank) | −$S^{\mathrm{IP}}$ (rank) | agg=$\Delta a$ | agg=$S^{\mathrm{IP}}$ | only $\Delta a$ | only $S^{\mathrm{IP}}$ | only $S^{\mathrm{Q\text{-}diff}}$ |
> |---|---|---|---|---|---|---|---|---|---|
> | LLaVA-1.5-7B | RefCOCO tA | **84.6** | 63.2 | 80.5 | 71.3 | 78.6 | 63.4 | 39.1 | 80.7 |
> | LLaVA-1.5-7B | RefCOCO+ tA | **77.4** | 62.3 | 73.1 | 63.8 | 73.4 | 55.2 | 42.2 | 71.0 |
> | LLaVA-1.5-7B | RefCOCOg t | **69.8** | 49.1 | 63.3 | 46.2 | 65.1 | 32.0 | 37.2 | 63.4 |
> | LLaVA-1.5-13B | RefCOCO tA | **82.8** | 75.1 | 79.1 | 58.5 | 71.0 | 43.6 | 32.9 | 78.6 |
> | LLaVA-1.5-13B | RefCOCO+ tA | **74.9** | 70.4 | 70.5 | 50.3 | 65.6 | 33.6 | 36.6 | 70.6 |
> | LLaVA-1.5-13B | RefCOCOg t | **71.3** | 65.0 | 70.1 | 39.4 | 60.9 | 27.2 | 34.0 | 67.7 |
> | BakLLaVA-v1 | RefCOCO tA | **80.0** | 66.1 | 76.4 | 62.2 | 69.2 | 58.8 | 20.6 | 77.5 |
> | BakLLaVA-v1 | RefCOCO+ tA | **71.3** | 58.1 | 70.3 | 52.3 | 63.0 | 50.5 | 17.1 | 65.3 |
> | BakLLaVA-v1 | RefCOCOg t | **64.3** | 54.9 | 60.1 | 39.9 | 52.8 | 33.7 | 16.4 | 61.3 |
> | Qwen2-VL-7B | RefCOCO tA | **68.0** | 58.6 | 63.0 | 28.4 | 43.4 | 10.7 | 31.0 | 31.9 |
> | Qwen2-VL-7B | RefCOCO+ tA | **65.9** | 55.5 | 64.1 | 26.2 | 45.5 | 11.6 | 33.7 | 31.3 |
> | Qwen2-VL-7B | RefCOCOg t | **52.0** | 47.0 | 49.4 | 20.1 | 35.2 | 9.2 | 30.2 | 24.6 |
> | **Average (12 settings)** | | **71.9** | 60.4 | 68.3 | 46.6 | 60.3 | 35.8 | 30.9 | 60.3 |

---

> ### Author Response · Authors · 2026-07-15
> **Response to Reviewer yFsZ (4)**
>
> **Q2. Contribution beyond top-$K$ head selection**
>
> Yes --- extracting spatial maps from text-to-image attention is established (ViTs, diffusion models, and Kang et al. for LVLMs), and we say so in Secs. 1--2. To make the delta unambiguous, here is a stage-by-stage accounting, which we will add verbatim to the revision:
>
> **Shared with prior work:** the LVLM interaction probe $q_{\mathrm{txt}}$ over image keys (Kang et al.); the generic smooth → binarize → extract-region post-processing skeleton (standard across attention-map grounding); the benchmarks and metrics.
>
> **New in this paper:** Our novelty is concentrated in two places, of which our controlled ablation (Appendix A.2) shows the first is the one carrying performance:
>
> 1. **The query-difference contrastive scoring** (rather than raw attention), which suppresses background activations and yields cleaner, more mask-like maps. Table 8 of the main paper quantifies this in-domain (**79.49%** vs. **22.52%** point accuracy), and ablation reproduces it out-of-domain (**+11.7** points, same heads).
>
> 2. **How the heads are identified** --- automatically, *per input*, via contrastive prompting with *zero* labeled data (vs. a fixed, batch-calibrated head set in prior work). We present this as a label-free convenience; it is competitive with, not superior to, their dataset-calibrated fixed heads on accuracy.
>
> Crucially, the new controls below show empirically that the contribution is not "top-$K$ selection" as a generic idea: replacing our contrastive scoring with per-sample *attention-magnitude* ranking --- i.e., the closest attention-statistics analogue --- collapses pointing accuracy from 71.9% to **15.0%** on average.
>
> Selection and prompt controls, pointing accuracy (%):
>
> | Model | Split | Full (ours) | Random-$k$ | Attn. magnitude | Neutral contrast |
> |---|---|---|---|---|---|
> | LLaVA-1.5-7B | RefCOCO tA | 84.6 | 73.9 | 23.2 | 78.3 |
> | LLaVA-1.5-7B | RefCOCO+ tA | 77.4 | 66.9 | 21.5 | 73.1 |
> | LLaVA-1.5-7B | RefCOCOg t | 69.8 | 60.2 | 28.9 | 67.1 |
> | LLaVA-1.5-13B | RefCOCO tA | 82.8 | 71.8 | 8.9 | 80.2 |
> | LLaVA-1.5-13B | RefCOCO+ tA | 74.9 | 66.2 | 9.1 | 73.1 |
> | LLaVA-1.5-13B | RefCOCOg t | 71.3 | 60.1 | 8.4 | 69.7 |
> | BakLLaVA-v1 | RefCOCO tA | 80.0 | 71.1 | 18.2 | 79.3 |
> | BakLLaVA-v1 | RefCOCO+ tA | 71.3 | 64.1 | 13.7 | 73.2 |
> | BakLLaVA-v1 | RefCOCOg t | 64.3 | 58.5 | 14.3 | 62.1 |
> | Qwen2-VL-7B | RefCOCO tA | 68.0 | 57.4 | 11.7 | 65.7 |
> | Qwen2-VL-7B | RefCOCO+ tA | 65.9 | 54.5 | 11.7 | 64.6 |
> | Qwen2-VL-7B | RefCOCOg t | 52.0 | 44.2 | 10.2 | 48.6 |
> | **Average (12 settings)** | | **71.9** | 62.4 | 15.0 | 69.6 |

---

### Review · Reviewer_GwGU · 2026-06-30

**Summary Of Contributions:**

This paper proposes a training-free visual grounding framework for frozen large vision-language models. The main idea is that only a small subset of attention heads in LVLMs encode useful spatial grounding signals. The authors discover these heads automatically using contrastive prompting between a target expression and a spatially opposed contrastive description. They define four complementary scoring functions: output-projection attribution, image-patch attribution, attention-difference score, and query-difference score. The top-ranked heads are then aggregated to produce localization heatmaps and bounding boxes. The method is evaluated on RefCOCO, RefCOCO+, RefCOCOg, and Flickr30K Entities using several LVLMs, including LLaVA-1.5-7B/13B, BakLLaVA, and Qwen2-VL.

**Audience:**

Yes

**Audience Explanation:**

The paper addresses a topic that would interest at least some of the TMLR audience, especially researchers working on vision-language models, visual grounding, interpretability, and training-free adaptation of large multimodal models. The idea that frozen LVLMs may already contain usable grounding information in only a few attention heads is interesting both practically and scientifically. It suggests that visual grounding can be partially extracted from existing LVLMs without fine-tuning, architectural changes, or labeled grounding data.

**Broader Impact Concerns:**

The method improves training-free visual grounding in LVLMs, which can have positive applications in assistive vision systems, image retrieval, robotics, and model interpretability. However, visual grounding systems can also be used in surveillance, person tracking, and automated monitoring, especially because the method does not require labeled grounding data or fine-tuning. The paper should briefly discuss this dual-use risk.

**Claims And Evidence:**

No

**Claims Explanation:**

I do not think all claims are fully supported by accurate and clear evidence. The paper makes relatively strong mechanistic claims, such as spatial grounding signals being “linearly separable” in the residual stream and specialized localization heads being discovered without supervision. These claims are plausible, but the evidence is mostly indirect through localization performance. The paper would need stronger causal or diagnostic experiments, such as head ablation, activation patching, linear probing, or controlled intervention studies, to convincingly establish these mechanistic claims.

**Requested Changes:**

(1) Clarify the role of the open-vocabulary detector in the final Acc@0.5 results.
(2) Provide stronger evidence for the mechanistic claims about linear separability and specialized heads.
(3) Add ablations for heuristic design choices.
(4) Strengthen comparison with Kang et al. and other head-selection methods.
(5) Explain model-dependent behavior and report runtime and computational cost.

---

> ### Author Response · Authors · 2026-07-15
> **Response to Reviewer GwGU (1)**
>
> We thank the reviewer for the careful and constructive review. We address the reviewer’s concerns below.
>
> **Q1. Role of the open-vocabulary detector in the final Acc@0.5 results**
>
> **The detector supplies box geometry, not localization.** Our contrastive heatmap lives on the LVLM's visual-token grid — $24\times24$ patches for LLaVA-style models, each covering a $14\times14$-pixel cell of the $336\times336$ input. This is fine enough to locate the object but too coarse for the tight boxes IoU $\geq 0.5$ requires: box edges can be off by up to a patch width, which alone pushes small or thin objects below 0.5. We therefore *snap* the heatmap's rough box to the YOLO-World proposal with maximum overlap. The detector never sees the referring expression or the ground truth; it only converts a location the LVLM has already chosen into a pixel-accurate box. Accordingly, we report point-in-box accuracy (fully detector-free, isolating the LVLM's localization signal) and Acc@0.5 (which additionally requires box geometry and thus uses detector fusion) — standard practice for training-free methods (GroundVLP).
>
> **Detector-only controls.** We evaluated the detector in isolation on the RefCOCO validation set under the identical proposal configuration (generic vocabulary; expression never seen), replacing our heatmap-based selection with expression-blind rules, plus one diagnostic that grants the detector the expression directly:
>
> | Selection rule | Acc@0.5 | mIoU |
> |---|---|---|
> | Oracle (max-IoU vs. GT; upper bound) | 95.8 | 0.885 |
> | Random proposal | 10.5 | 0.131 |
> | Largest-area proposal | 25.9 | 0.316 |
> | Highest-confidence proposal | 28.3 | 0.308 |
> | Detector + referring expression (CLIP matching) | 34.6 | 0.356 |
> | **Ours (heatmap-selected; paper Table 1)** | **70.2** | — |
>
> The proposal set almost always contains a correct box (oracle 95.8%), yet no expression-blind rule recovers it (≤28.3%), and even with the expression the detector reaches only 34.6%, versus our 70.2% and detector-free point accuracy of 79.49% (LLaVA-1.5-7B, Table 2). The LVLM localizes; the detector converts an already-identified region into a pixel-accurate box.
>
> **Q5. Model-dependent behavior; runtime and computational cost**
>
> **(a) Model-dependent behavior.** In brief: the LLaVA-family models combine strong spatial representations with concentrated grounding circuits; BakLLaVA-v1 has the representation but a diffuse circuit; Qwen2-VL has both, but needs the full score combination to access them.
>
> 1. **All four models know where the object is.** Linear probes read the referent's location from every model's hidden states at 63–69% on average and up to 74%. The differences between models are therefore *not* caused by missing spatial representations.
> 2. **What differs is how concentrated that knowledge is in a few heads.** Patching the top-10 heads recovers 23–28% of the flipped decision margin in LLaVA-1.5-7B but only 2–7% in BakLLaVA-v1, and head ablation shows the same gap.
> 3. **Qwen2-VL fails for a different reason: score geometry, not representation.** Its probes are the strongest of the four, yet the raw query-difference score alone collapses on it (24.6–31.9% pointing) — plausibly because its M-RoPE positions and dynamic-resolution tokens alter the query–key geometry. The four-score combination recovers it to 52–68%, evidencing that the combined criterion is a robustness mechanism, not an arbitrary design choice.
>
> **(b) Runtime and computational cost.** Per-image runtime and memory on a single GPU (4-bit NF4-quantized weights):
>
> | Model | Descriptor extr. | Pass A | Pass B | Scores | YOLO | **Total (s/img)** | Peak mem. (GB) |
> |---|---|---|---|---|---|---|---|
> | LLaVA-1.5-7B | 0.255 | 0.253 | 0.253 | 0.011 | 0.020 | **0.79** | 6.99 |
> | LLaVA-1.5-13B | 0.398 | 0.404 | 0.405 | 0.014 | 0.020 | **1.24** | 11.83 |
> | BakLLaVA-v1 | 0.249 | 0.250 | 0.251 | 0.013 | 0.020 | **0.78** | 6.58 |
> | Qwen2-VL-7B | 0.315 | 0.312 | 0.317 | 0.011 | 0.019 | **0.97** | 15.42 |
>
> Most importantly, this is the *entire* cost: there is no fine-tuning, no labeled calibration set, and no per-dataset head-selection stage.
>
> **Broader Impact**
>
> We have added the following Broader Impact paragraph as new Section 5.6 to the revised paper:
>
> > Our method improves training-free visual grounding in frozen LVLMs, benefiting assistive vision, image retrieval, robotics, and interpretability. Like all grounding technology it is dual-use — it could be repurposed for surveillance or tracking, and its training-free nature lowers the barrier to such deployment. Two factors limit this risk: the method surfaces representations the frozen LVLM and off-the-shelf detectors already encode rather than creating new capability, and its accuracy remains well below the fine-tuned systems high-stakes deployments would realistically use. Downstream use must comply with applicable privacy regulations; we view making LVLM grounding transparent and auditable as a net contribution to responsible AI.

---

> ### Author Response · Authors · 2026-07-15
> **Response to Reviewer GwGU (2)**
>
> **Q2. Stronger evidence for linear separability and specialized heads**
>
> Before presenting our new experiments, we note that neither claim is exotic. Linear decodability of semantic features from transformer residual streams is among the most consistently replicated findings in Interpretability research (Alain \& Bengio, 2016; Park et al., 2023), and most directly relevant to our setting. Merullo et al. (2022) show that image representations map into text space via a single linear transformation, indicating that visual and textual features share an approximately linear geometry in LVLMS. Likewise, functional specialization of a small subset of attention heads is a recurring result across architectures and tasks (Voita et al 2019).
>
>  To be precise about scope: we do not claim novelty for these general phenomena — they are inherited from prior work. Our claim, tested directly below, is that spatial grounding in frozen LVLMs is a further instance: the grounding signal is linearly decodable from the residual stream, and concentrated in a small head subset that is causally responsible for grounding behavior. We ran all three suggested diagnostics — (a) linear probing with a shuffled-label control, (b) causal head ablation, (c) activation patching — providing converging correlational *and* causal evidence.
>
> **(a) Linear probes.** For each layer, a linear classifier is trained on the frozen residual stream at the final text-token position to predict the referent's quadrant (plus a ridge regression for its center coordinates), with a shuffled-label control; probe information is never used by our method.
>
> | Model | Split | $L^*$ (of $N$) | Probe acc. | Shuffled acc. | Gap | Peak center corr. |
> |---|---|---|---|---|---|---|
> | LLaVA-1.5-7B | RefCOCO testA | 19 / 32 | **72.2** | 29.2 | +43.0 | 0.79 |
> | LLaVA-1.5-7B | RefCOCO+ testA | 15 / 32 | **62.7** | 29.5 | +33.2 | 0.65 |
> | LLaVA-1.5-7B | RefCOCOg test | 14 / 32 | **57.5** | 28.8 | +28.7 | 0.63 |
> | LLaVA-1.5-13B | RefCOCO testA | 22 / 40 | **74.2** | 28.3 | +45.9 | 0.81 |
> | LLaVA-1.5-13B | RefCOCO+ testA | 16 / 40 | **65.8** | 26.0 | +39.8 | 0.69 |
> | LLaVA-1.5-13B | RefCOCOg test | 20 / 40 | **65.5** | 26.3 | +39.2 | 0.68 |
> | BakLLaVA-v1 | RefCOCO testA | 18 / 32 | **68.5** | 29.7 | +38.8 | 0.76 |
> | BakLLaVA-v1 | RefCOCO+ testA | 19 / 32 | **62.9** | 27.5 | +35.4 | 0.61 |
> | BakLLaVA-v1 | RefCOCOg test | 19 / 32 | **58.8** | 28.0 | +30.8 | 0.63 |
> | Qwen2-VL-7B | RefCOCO testA | 19 / 28 | **74.2** | 31.7 | +42.5 | 0.82 |
> | Qwen2-VL-7B | RefCOCO+ testA | 18 / 28 | **68.3** | 27.5 | +40.8 | 0.74 |
> | Qwen2-VL-7B | RefCOCOg test | 21 / 28 | **63.7** | 27.8 | +35.8 | 0.74 |
>
> The probe reads location far above chance in every model (up to 74.2% quadrant accuracy; center correlation up to 0.82) while the shuffled control stays at chance — spatial information is linearly decodable. Accuracy peaks at middle depth, mirroring the layer distribution of our discovered heads (paper Table 5): two independent measurements localize the same computational stage. Following the reviewer's concern, we will revise "linearly separable" to the more precise "*linearly decodable* from the residual stream."
>
> **(b) Causal head ablation.** We use a left/right diagnostic that reads the *model's own* logit margin (not our heatmap pipeline, so the test cannot be circular), and mean-ablate $K{=}10$ heads at the final token: our selected heads vs. bottom and random controls. $\Delta$margin = drop in the left/right margin after ablation (larger = those heads matter more).
>
> | Model | Split | Clean margin | $\Delta$ Selected | $\Delta$ Bottom | $\Delta$ Random | Sel.−rand. mean [95% CI] |
> |---|---|---|---|---|---|---|
> | LLaVA-1.5-7B | RefCOCO testA | 1.21 | **0.393** | 0.345 | −0.002 | 0.395 [0.360, 0.432] |
> | LLaVA-1.5-7B | RefCOCO+ testA | 1.98 | **0.514** | −0.001 | −0.002 | 0.517 [0.465, 0.568] |
> | LLaVA-1.5-7B | RefCOCOg test | 1.05 | **0.292** | −0.002 | 0.002 | 0.289 [0.256, 0.323] |
> | LLaVA-1.5-13B | RefCOCO testA | 1.68 | **0.278** | 0.001 | −0.028 | 0.306 [0.282, 0.329] |
> | LLaVA-1.5-13B | RefCOCO+ testA | 1.54 | **0.221** | 0.005 | −0.018 | 0.240 [0.215, 0.264] |
> | LLaVA-1.5-13B | RefCOCOg test | 1.37 | **0.303** | −0.001 | −0.018 | 0.322 [0.288, 0.357] |
> | BakLLaVA-v1 | RefCOCO testA | 2.28 | **0.295** | 0.096 | −0.072 | 0.367 [0.313, 0.421] |
> | BakLLaVA-v1 | RefCOCO+ testA | 2.58 | 0.049 | 0.105 | −0.026 | 0.075 [0.052, 0.098] |
> | BakLLaVA-v1 | RefCOCOg test | 1.67 | **0.071** | 0.005 | −0.025 | 0.096 [0.073, 0.119] |
> | Qwen2-VL-7B | RefCOCO testA | 2.63 | **0.221** | −0.030 | −0.010 | 0.231 [0.219, 0.242] |
> | Qwen2-VL-7B | RefCOCO+ testA | 2.57 | **0.187** | 0.028 | −0.015 | 0.202 [0.191, 0.214] |
> | Qwen2-VL-7B | RefCOCOg test | 2.00 | **0.160** | −0.003 | −0.005 | 0.165 [0.158, 0.172] |

---

> > ### Author Response · Authors · 2026-07-15
> > **Q2. Stronger evidence for linear separability and specialized heads- Continue**
> >
> > **(c) Activation patching.** We flip the image horizontally (moving the referent to the other side, flipping the margin), then patch the $K{=}10$ heads' clean final-token activations into the flipped run and measure restoration, against random/bottom controls. Values in logits; negative "Flipped" margin = the model correctly follows the flip.
> >
> > | Model | Split | Clean | Flipped | Rest. Selected | Rest. Random | Rest. Bottom | Sel.−rand. mean [95% CI] |
> > |---|---|---|---|---|---|---|---|
> > | LLaVA-1.5-7B | RefCOCO testA | +1.49 | −1.04 | **+0.59** | −0.00 | +0.46 | 0.594 [0.560, 0.626] |
> > | LLaVA-1.5-7B | RefCOCO+ testA | +1.96 | −1.88 | **+0.88** | −0.01 | −0.00 | 0.883 [0.848, 0.918] |
> > | LLaVA-1.5-7B | RefCOCOg test | +1.00 | −0.82 | **+0.51** | +0.00 | −0.00 | 0.507 [0.491, 0.523] |
> > | LLaVA-1.5-13B | RefCOCO testA | +1.68 | −0.32 | **+0.30** | −0.01 | −0.00 | 0.308 [0.268, 0.349] |
> > | LLaVA-1.5-13B | RefCOCO+ testA | +1.54 | −1.41 | **+0.42** | −0.02 | −0.00 | 0.437 [0.397, 0.477] |
> > | LLaVA-1.5-13B | RefCOCOg test | +1.37 | −0.96 | **+0.50** | −0.01 | +0.00 | 0.510 [0.453, 0.566] |
> > | BakLLaVA-v1 | RefCOCO testA | +2.28 | −0.89 | **+0.23** | −0.03 | +0.03 | 0.256 [0.215, 0.298] |
> > | BakLLaVA-v1 | RefCOCO+ testA | +2.58 | −2.24 | **+0.09** | −0.05 | +0.04 | 0.134 [0.117, 0.150] |
> > | BakLLaVA-v1 | RefCOCOg test | +1.67 | −1.61 | **+0.09** | −0.03 | +0.00 | 0.123 [0.106, 0.141] |
> > | Qwen2-VL-7B | RefCOCO testA | +2.63 | −0.66 | **+0.23** | +0.00 | −0.02 | 0.232 [0.220, 0.245] |
> > | Qwen2-VL-7B | RefCOCO+ testA | +2.57 | −2.51 | **+0.36** | +0.00 | −0.04 | 0.355 [0.343, 0.368] |
> > | Qwen2-VL-7B | RefCOCOg test | +2.00 | −1.54 | **+0.25** | −0.00 | −0.00 | 0.252 [0.245, 0.260] |
> >
> > Only patching our selected heads pulls the answer back — up to $+0.88$ logits (23–28% of the clean-vs-flipped gap on LLaVA-1.5-7B) — while random heads restore ≈0 in all 12 settings ($p<0.001$; every selected-vs-random CI excludes zero). One transparent exception: bottom heads also restore $+0.46$ on LLaVA-1.5-7B/RefCOCO testA, indicating some redundant spatial signal in low-ranked heads on that split; selected heads still restore the most.
> >
> > Together, (a)–(c) establish both claims: the grounding signal is linearly decodable from the residual stream, and the heads our unsupervised procedure discovers are causally responsible for grounding behavior.

---

> ### Author Response · Authors · 2026-07-15
> **Response to Reviewer GwGU (3)**
>
> **Q3. Ablations for heuristic design choices**
>
> **(a) Factorial ablation of the four scoring functions.** The scores play two roles: *ranking* heads via $r_{\ell,h}=\sum_p \Delta a_{\ell,h}[p]\cdot s^{\mathrm{IP}}_{\ell,h,p}$ (Eq. 10) and *aggregating* the heatmap from $S^{\mathrm{Q\text{-}diff}}$ (Eq. 13); we drop each factor from the ranking, swap the aggregation signal, and run each score alone. Three findings: (i) **both ranking factors matter** — removing $\Delta a$ costs −11.5 points on average and removing $S^{\mathrm{IP}}$ costs −3.6; the Eq. 10 product beats either factor alone in every setting; (ii) **aggregation must use $S^{\mathrm{Q\text{-}diff}}$** — aggregating $\Delta a$ maps instead collapses the heatmap (Acc@0.5 ≈ 0); (iii) **no single score is architecture-robust** — the best one, only $S^{\mathrm{Q\text{-}diff}}$, trails in every LLaVA-style setting (by 2.5–6.4 points) and collapses on Qwen2-VL (24.6–31.9 vs. 52.0–68.0), plausibly because M-RoPE and dynamic-resolution processing change its query–key geometry. The full combination is best in all 12 settings; it is a robustness mechanism, not an arbitrary choice.
>
> Pointing accuracy (%):
>
> | Model | Split | Full | −$\Delta a$ (rank) | −$S^{\mathrm{IP}}$ (rank) | agg=$\Delta a$ | agg=$S^{\mathrm{IP}}$ | only $\Delta a$ | only $S^{\mathrm{IP}}$ | only $S^{\mathrm{Q\text{-}diff}}$ |
> |---|---|---|---|---|---|---|---|---|---|
> | LLaVA-1.5-7B | RefCOCO tA | 84.6 | 63.2 | 80.5 | 71.3 | 78.6 | 63.4 | 39.1 | 80.7 |
> | LLaVA-1.5-7B | RefCOCO+ tA | 77.4 | 62.3 | 73.1 | 63.8 | 73.4 | 55.2 | 42.2 | 71.0 |
> | LLaVA-1.5-7B | RefCOCOg t | 69.8 | 49.1 | 63.3 | 46.2 | 65.1 | 32.0 | 37.2 | 63.4 |
> | LLaVA-1.5-13B | RefCOCO tA | 82.8 | 75.1 | 79.1 | 58.5 | 71.0 | 43.6 | 32.9 | 78.6 |
> | LLaVA-1.5-13B | RefCOCO+ tA | 74.9 | 70.4 | 70.5 | 50.3 | 65.6 | 33.6 | 36.6 | 70.6 |
> | LLaVA-1.5-13B | RefCOCOg t | 71.3 | 65.0 | 70.1 | 39.4 | 60.9 | 27.2 | 34.0 | 67.7 |
> | BakLLaVA-v1 | RefCOCO tA | 80.0 | 66.1 | 76.4 | 62.2 | 69.2 | 58.8 | 20.6 | 77.5 |
> | BakLLaVA-v1 | RefCOCO+ tA | 71.3 | 58.1 | 70.3 | 52.3 | 63.0 | 50.5 | 17.1 | 65.3 |
> | BakLLaVA-v1 | RefCOCOg t | 64.3 | 54.9 | 60.1 | 39.9 | 52.8 | 33.7 | 16.4 | 61.3 |
> | Qwen2-VL-7B | RefCOCO tA | 68.0 | 58.6 | 63.0 | 28.4 | 43.4 | 10.7 | 31.0 | 31.9 |
> | Qwen2-VL-7B | RefCOCO+ tA | 65.9 | 55.5 | 64.1 | 26.2 | 45.5 | 11.6 | 33.7 | 31.3 |
> | Qwen2-VL-7B | RefCOCOg t | 52.0 | 47.0 | 49.4 | 20.1 | 35.2 | 9.2 | 30.2 | 24.6 |
> | **Average (12 settings)** | | **71.9** | 60.4 | 68.3 | 46.6 | 60.3 | 35.8 | 30.9 | 60.3 |
>
> **(b) Head-selection and contrastive-prompt controls.** Three controls: *Random-k* (10 random heads, our aggregation kept), *Attention magnitude* (rank and aggregate by raw target-pass attention; no contrastive signal), and *Neutral contrast* (the spatially-opposed description replaced by a generic "Describe the overall scene" prompt). Random heads lose 9.5 points on average and raw attention magnitude collapses to 15.0% — **contrastive scoring, not attention mass, is what finds the localization heads**. Neutral contrast loses 2.3 points and is worse in 11 of 12 settings — **the spatial flip-mapping of Sec. 4.1 contributes beyond merely adding a second forward pass**.
>
> Pointing accuracy (%):
>
> | Model | Split | Full (ours) | Random-$k$ | Attn. magnitude | Neutral contrast |
> |---|---|---|---|---|---|
> | LLaVA-1.5-7B | RefCOCO tA | 84.6 | 73.9 | 23.2 | 78.3 |
> | LLaVA-1.5-7B | RefCOCO+ tA | 77.4 | 66.9 | 21.5 | 73.1 |
> | LLaVA-1.5-7B | RefCOCOg t | 69.8 | 60.2 | 28.9 | 67.1 |
> | LLaVA-1.5-13B | RefCOCO tA | 82.8 | 71.8 | 8.9 | 80.2 |
> | LLaVA-1.5-13B | RefCOCO+ tA | 74.9 | 66.2 | 9.1 | 73.1 |
> | LLaVA-1.5-13B | RefCOCOg t | 71.3 | 60.1 | 8.4 | 69.7 |
> | BakLLaVA-v1 | RefCOCO tA | 80.0 | 71.1 | 18.2 | 79.3 |
> | BakLLaVA-v1 | RefCOCO+ tA | 71.3 | 64.1 | 13.7 | 73.2 |
> | BakLLaVA-v1 | RefCOCOg t | 64.3 | 58.5 | 14.3 | 62.1 |
> | Qwen2-VL-7B | RefCOCO tA | 68.0 | 57.4 | 11.7 | 65.7 |
> | Qwen2-VL-7B | RefCOCO+ tA | 65.9 | 54.5 | 11.7 | 64.6 |
> | Qwen2-VL-7B | RefCOCOg t | 52.0 | 44.2 | 10.2 | 48.6 |
> | **Average (12 settings)** | | **71.9** | 62.4 | 15.0 | 69.6 |

---

> ### Author Response · Authors · 2026-07-15
> **Response to Reviewer GwGU (4)**
>
> **Q4. Strengthen comparison with Kang et al. and other head-selection methods**
>
> We have expanded this section in three ways:
>
> 1. **Setting-level positioning.** Kang et al. (2025) and our work answer the same question --- which heads carry grounding? --- under different information budgets. They compute attention-sum statistics over 1,000 labeled RefCOCO samples to calibrate a threshold $\tau$ and head-selection frequencies, fixing a per-model head set ($k{=}3$) reused for all test inputs; we discover heads per (image, expression) at inference time with no labeled data or calibration. The table below positions this alongside earlier head-analysis work and F-LMM.
>
> 2. **Accuracy under each budget.** With calibrated fixed heads, Kang et al. are more accurate (87.2 vs. our 69.35 Acc@0.5, LLaVA-1.5-13B, RefCOCO val); we do not claim otherwise. Under a matched zero-label budget, however, their own ablation (their Table 5) shows applying their criteria per sample drops performance to 67.4 --- below our 69.35 --- while still retaining the calibrated $\tau$. As they note, spatial entropy finds heads that are *localized* but not necessarily *text-referred*; only their dataset-level frequency statistic filters distractor-locked heads, whereas our contrastive dual-pass supplies text-referredness per sample by construction. One asymmetry: their boxes come from a convex hull of the attention map, ours from snapping to a training-free detector that sees neither the expression nor labels --- so the budget stays matched, and our detector-only controls show the detector alone reaches $\leq$34.6 Acc@0.5, ruling out the refinement step as the explanation.
>
> 3. **Causal validation, which no prior head-selection method provides.** Kang et al. validate their heads correlationally ($\rho > 0.7$); our ablation and activation-patching experiments (response to Q2) show our discovered heads are *causally responsible* for grounding behavior. The criterion is causally motivated by construction: contrastive prompting induces two spatially opposed readings, and heads are ranked by their measured contribution to the output difference (Eq. 10). The same heads' query--key geometry yields our heatmaps (Eqs. 12--13), so head identification and heatmap extraction are two readouts of one circuit, not independent heuristics. Robustness also differs: their head-count ablation degrades sharply past its optimum (67.1 at $k{=}3$ → 58.9 at $k{=}5$, their Table 4), while ours is nearly flat from $k{=}10$ to $k{=}20$ (71.2 → 70.7, paper Table 4), suggesting contrastive scoring surfaces a deeper pool of usable heads.
>
> Positioning ("Granularity" = whether the head set adapts to each input; "Causal validation" = verified via intervention rather than correlation alone):
>
> | Method | Selection signal | Labels / training | Granularity | New domain needs | Causal validation |
> |---|---|---|---|---|---|
> | Voita et al. (2019); Clark et al. (2019) | task-supervised pruning / probing | task labels | fixed | re-analysis | partial (pruning) |
> | F-LMM (Wu et al., 2025) | frozen attention + trained refiner | grounding data (training) | fixed (learned) | retraining refiner | --- |
> | Kang et al. (2025) | attention sum + spatial entropy stats. | 1,000 samples + $\tau$ | fixed per model | re-calibration (1,000 samples) | --- |
> | Kang et al., greedy (their Tab. 5) | same, applied per sample | $\tau$ still calibrated | per-sample | re-calibration ($\tau$) | --- |
> | **Ours** | **contrastive residual / attention / query analysis** | **none** | **per-sample** | **none** | **ablation + patching (Q2)** |
>
> **References**- Kang, S., Kim, J., Kim, J., Hwang, S. J. "Your Large Vision-Language Model Only Needs A Few Attention Heads For Visual Grounding." CVPR 2025.- Wu, S., Jin, S., Zhang, W., Xu, L., Liu, W., Li, W., Loy, C. C. "F-LMM: Grounding Frozen Large Multimodal Models." CVPR 2025.- Voita, E., Talbot, D., Moiseev, F., Sennrich, R., Titov, I. "Analyzing Multi-Head Self-Attention: Specialized Heads Do the Heavy Lifting, the Rest Can Be Pruned." ACL 2019.- Clark, K., Khandelwal, U., Levy, O., Manning, C. D. "What Does BERT Look At? An Analysis of BERT's Attention." BlackboxNLP @ ACL 2019.

---

### Author Response · Authors · 2026-07-15
**Note to the Action Editor Regarding Review yFsZ**

We would like to flag a concern with Reviewer yFsZ's review.

**1. The central claim of the review is factually incorrect.**
The reviewer recommends rejection on the grounds that our paper *"did not explicitly tell the connections"* to Kang et al. (CVPR 2025) and calls this *"very unethical."* However, the submission **cites, describes, and empirically compares against Kang et al. in four places**:
- **Sec. 2 (Related Work)** — a dedicated paragraph describing their method and its reliance on 1,000 labeled RefCOCO samples;
- **Sec. 5.1 (Baselines)** — Kang et al. listed as a dedicated baseline category;
- **Table 1** — their results reported in a clearly labeled tier († 1,000 labeled samples required for head selection), visually separated from our zero-shot tier;
- **Sec. 5.5** — an explicit mechanistic contrast with their approach.

Full quotations are provided in our rebuttal.

**2. The objection rests on a paper the manuscript already addresses.**
The reviewer states they are *"not very familiar"* with visual grounding, and their sole substantive objection rests on a related paper found post hoc — a relationship the manuscript already discusses transparently, including the key setting difference (**1,000 labeled samples vs. zero** for head selection).

**3. Request.**
We take ethics allegations seriously and believe this one is unsupported by the text of the submission. We respectfully ask that this be taken into account when weighing the review, and we remain happy to provide any further clarification.

---

### Author Response · Authors · 2026-07-15
**General Response to All Reviewers**

**Summary of revisions**

We thank all reviewers for their feedback. A revised manuscript is uploaded with all changes highlighted in yellow, including a new appendix:

- **App. A.1:** linear probes, head ablation, activation patching (linear decodability + causal evidence for grounding heads)
- **App. A.2:** scoring-function ablations and control experiments
- **App. A.3:** role of the open-vocabulary detector
- **App. A.4:** model-dependent behavior and runtime/cost
- **App. A.5:** effect of the number of selected heads $k$